# Corticotropin-releasing hormone modulates NREM sleep consolidation through the thalamic reticular nucleus

Loredana Cumpana[1,2], Olivia Zanoletti[1,2], Dinesh Kankanamge [3], Bryan Copits [3], Carmen Sandi [1,2] ✉ & Simone Astori [1,2] ✉

Corticotropin-releasing hormone (CRH) is a peptide associated with stress and anxiety that acts as a potent modulator throughout the nervous system. The thalamic reticular nucleus (TRN) displays high expression of the CRH receptor 1 (CRHR1), but whether CRH modulates key TRN functions, such as sleep spindle rhythmogenesis, remained unexplored. Combining polysomnographic and photometric recordings in mice, we show that CRH release in TRN during non-rapid-eye movement sleep (NREMS) oscillates with a ~50-s periodicity, anti-correlating with sleep spindle dynamics. Optogenetic manipulations of CRH release in TRN modulated NREMS fragmentation through microarousals with corresponding changes in sigma and delta power. In ex-vivo recordings, CRHR1 activation decreased the propensity of TRN neurons to fire calcium bursts. CRHR1 knockdown in parvalbumin TRN neurons prevented the effects of CRH on NREMS and TRN bursting. Thus, CRHR1 impacts NREMS by modulating thalamic excitability, providing a potential target to stabilize sleep impairments associated with stress and anxiety.

Disturbed and fragmented sleep has significant detrimental consequences not only on cognitive function during wakefulness but also increases the risk of developing psychiatric and various other secondary health conditions[1–3]. Several arousal-related neuromodulators, such as noradrenaline[4–7], serotonin[8], acetylcholine[9], and orexin[10] have been shown to modulate the sleep-wake cycle and sleep consolidation, in particular through their association with the occurrence of brief awakenings that include microarousals (MAs). MAs in mice are commonly defined as transient periods of EEG activation concomitant with EMG activity[11,12], and their excessive occurrence has been linked to increased acute stress and anxiety[2,5,13]. Corticotropin-releasing hormone (CRH), the main stress- and anxiety-related peptide, is widely released throughout the central nervous system with diverse functions[14,15]. CRH receptor 1 (CRHR1), the main receptor activated by CRH, has been reported to be highly expressed in the thalamic

reticular nucleus (TRN)[16], which is part of the thalamocortical loop, the main pattern generator for non-rapid eye movement sleep (NREMS) oscillations[17]. However, despite evidence linking stress and anxiety to NREMS fragmentation[2,18] and the expression of CRHR1 in TRN neurons[16], the specific role of CRH in modulating NREMS through its action in TRN neurons remains unexplored.

During NREMS, thalamocortical oscillations ensure sleep continuity, facilitating memory consolidation and restorative functions of sleep, while maintaining sensory arousability[17,19–21]. Among these oscillations, sleep spindles, initiated by the rhythmic bursting of TRN neurons and entrained within the thalamocortical loop[22,23], play a key role in memory consolidation, working in concert with slow oscillations and hippocampal ripples[24,25]. Throughout NREMS, spindles cluster within 50-s intervals that manifest as an infraslow oscillation (ISO) (0.02 Hz) in the EEG sigma power band (10–15 Hz)[26], which serves

[1]Laboratory of Behavioral Genetics, Brain Mind Institute, School of Life Sciences, Ecole Polytechnique Fédérale de Lausanne, Lausanne, Switzerland. [2]Synapsy Center for Neuroscience and Mental Health Research, School of Life Sciences, Ecole Polytechnique Fédérale de Lausanne, Lausanne, Switzerland. [3]Department of Anesthesiology, Washington University Pain Center, Washington University School of Medicine, St Louis, MO, USA. ✉e-mail: carmen.sandi@epfl.ch; simone.astori@epfl.ch

as a proxy for spindle density. The ISO of sigma has become an important and reliable marker for NREMS continuity and sensory arousability in mice[17,27] and humans[28], with its rising and falling phases corresponding to the alternating continuity and fragility substates of NREMS[17,26]. Notably, NREMS fragility periods have a higher probability for the occurrence of MAs[26,27], which are considered a hallmark of the reversibility and preservation of sensory arousability in sleep[11]. Higher densities of MAs are also a main marker of poor sleep quality, as they may induce NREMS fragmentation, leading to detrimental consequences on cognitive performance, memory consolidation, and mood regulation[18,29,30].

The preservation of arousal-related neuromodulation during NREMS at a low and intermittent level is emerging as a mediator for the increased vulnerability of NREMS fragility substates to MAs[4,31,32]. Particularly, recent studies have shown that noradrenaline is released in the TRN during NREMS, with infraslow fluctuations that anti-correlate with sigma power, contributing to the organization of sleep spindles within 50-s periods[32], and reaching its highest levels during MAs[5]. Whether other neuromodulators, such as CRH, exhibit a similar oscillatory pattern during NREMS and share functional similarities with noradrenaline remains an open question. This question can now be addressed with the development of fluorescent sensors that enable the monitoring of endogenous release of such molecules in the brain across the sleep-wake cycle[33,34].

Given the association between arousal-related neuromodulators and NREMS vulnerability to MAs through TRN excitability[32], we aimed here to investigate whether CRH release in TRN modulates NREMS

fragmentation and affects TRN-dependent oscillations, such as sleep spindles.

## Results

### CRHR1 mRNA is highly expressed in parvalbumin-positive TRN neurons

To investigate the potential role of CRH in modulating NREMS through the TRN, we quantified the CRHR1 mRNA expression in TRN neuronal subtypes using RNAscope fluorescent in situ hybridization. We observed a pronounced expression of CRHR1 mRNA throughout the TRN, significantly exceeding expression levels in regions such as the hippocampal CA3 and the basolateral amygdala (BLA), which have been reported to exhibit moderate to low CRHR1 mRNA expression[16] (Fig. 1a–e). Furthermore, the CRHR1 expression in TRN was comparable between male and female mice (Fig. 1f), indicating no sex-related differences. Importantly, parvalbumin-positive (PV+) TRN neurons showed substantially higher CRHR1 mRNA expression compared to somatostatin-positive (SOM+) neurons (Fig. 1g–i). Given that PV+ neurons are crucial for spindle generation[35–37], this suggests that these neurons could be particularly responsive to CRH, with a potential role in modulating NREMS.

### CRH-releasing afferents in the TRN originate from multiple central CRH sources

To identify the origin of the CRH input to TRN, we used the CRH-IRES-Cre transgenic mouse line[38], which constitutively expresses Cre recombinase under the CRH promoter. In mice injected with a

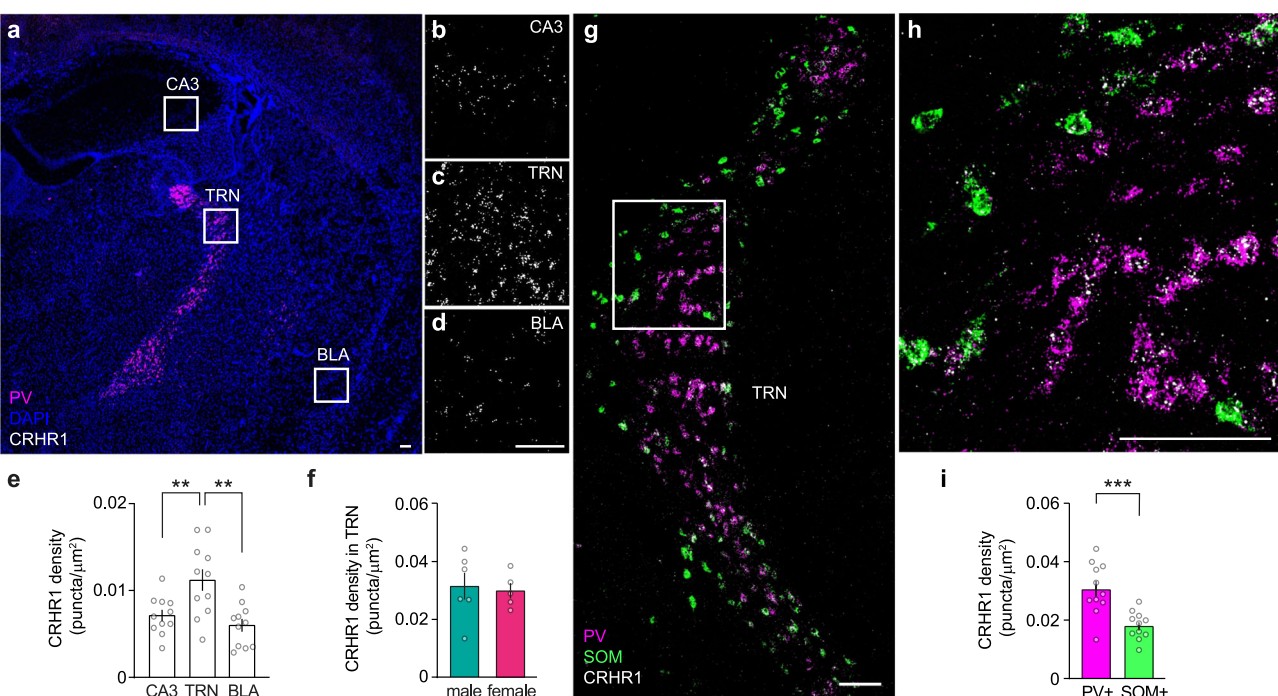

**Fig. 1 | CRHR1 mRNA is highly expressed in parvalbumin-positive TRN neurons.**
**a** Example image overview of RNAscope in situ mRNA hybridization stained for DAPI (blue), with parvalbumin (PV–magenta) highlighting the TRN and white squares indicating the corresponding regions shown as close-ups in hippocampal CA3 (**b**), TRN (**c**), and basolateral amygdala (BLA, **d**), where CRHR1 mRNA can be seen as white puncta. Quantification from independent observations is provided in (**e**). **e** TRN has significantly higher levels of CRHR1 mRNA compared to CA3 and BLA (n = 11). **f** No significant difference was found between males (n = 6) and females (n = 5) in CRHR1 mRNA expression in TRN. **g** Example image of in situ mRNA hybridization for parvalbumin (PV–magenta), somatostatin (SOM–green), and CRHR1 (white) mRNA labeling in TRN, showing the expected anatomical

segregation into a PV+ core and SOM+ shell in the sensory sector. **h** Close-up of TRN, exemplifying a higher density of CRHR1 mRNA puncta in PV+ neurons compared to SOM+ neurons. **i** CRHR1 mRNA has a significantly higher expression in PV+ TRN neurons. Each point is the average density of CRHR1 mRNA in PV+ or SOM+ only TRN nuclei detected with DAPI of a mouse (n = 11). Data are represented as mean ± SEM. All scale bars are 100 μm. Statistical analysis was performed by **e** repeated measures one-way ANOVA with Holm–Šídák's multiple comparisons test and **f**–**i** two-tailed unpaired t-test, with **p < 0.01, ***p < 0.001. For additional statistical information, see Supplementary Table 1. Source data are provided as a Source data file.

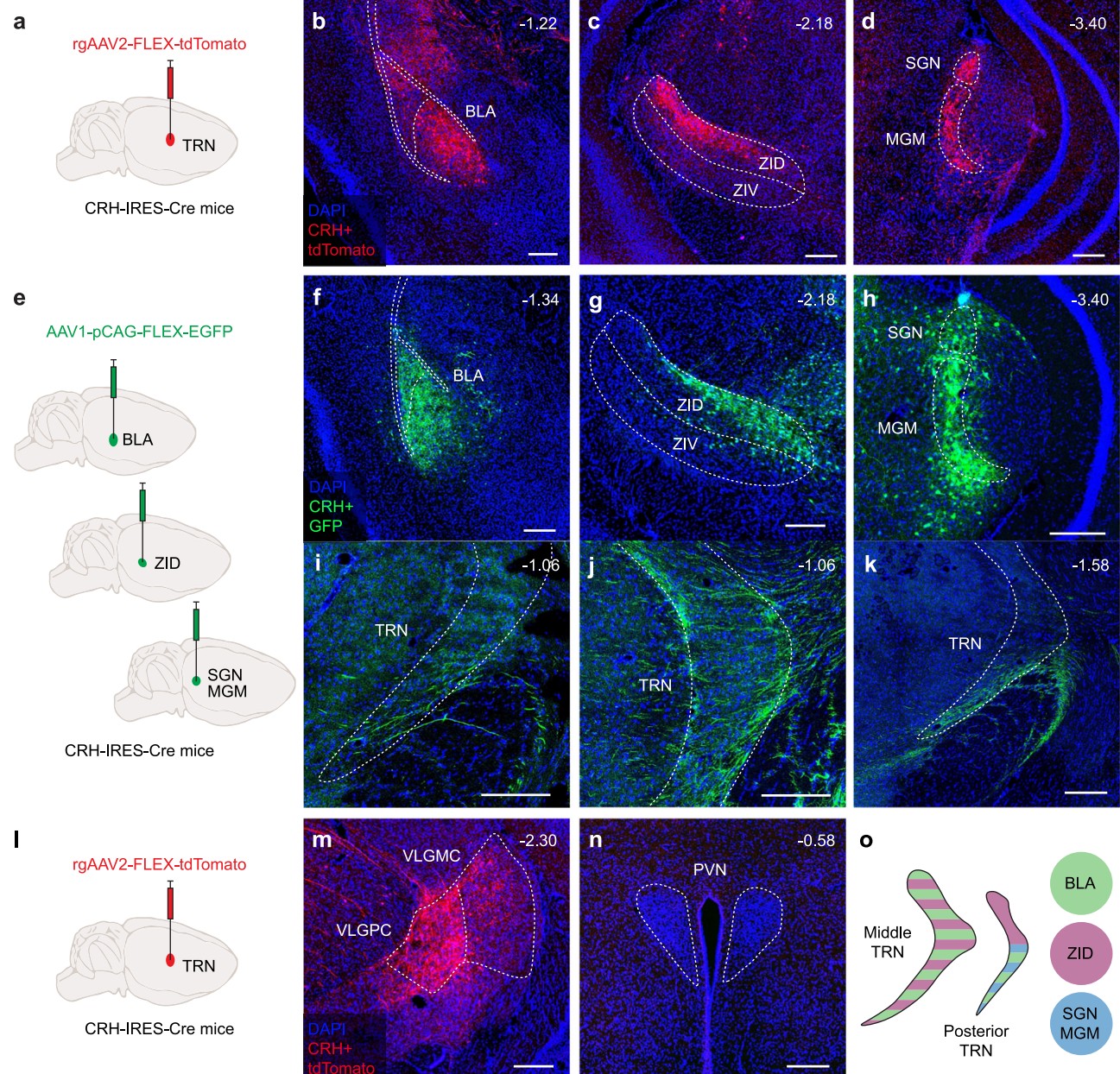

**Fig. 2 | CRH-releasing afferents in the TRN originate from multiple central CRH sources. a** Schematic of the retrograde tracing approach: a Cre-dependent tdTomato-expressing virus was injected into the sensory TRN to label CRH-releasing neurons projecting to this region. Representative images showing retrogradely labeled CRH+ neurons (tdTomato, red) in several brain regions projecting to the TRN, including the basolateral amygdala (BLA, **b**), dorsal zona incerta (ZID, **c**), superior geniculate nucleus (SGN, **d**), and medial division of the medial geniculate nucleus (MGM, **d**). **e** Schematic of the anterograde tracing strategy to anatomically confirm projections from CRH-expressing neurons (GFP-labeled, green) in the identified regions. Examples of injection sites for anterograde tracing in BLA (**f**), ZID (**g**), SGN, and MGM (**h**). Anterogradely labeled CRH+ axons from BLA (**i**), ZID (**j**), and SGN/MGM (**k**) were observed contacting distinct regions of the TRN.

**l, m** Retrograde tracing revealed CRH+ neurons in the ventrolateral parvocellular division of the geniculate nucleus (VLGPC) as a candidate TRN-projecting population; however, this projection was not confirmed with anterograde tracing. **n** No retrogradely labeled CRH+ neurons were found in the paraventricular nucleus of the hypothalamus (PVN), suggesting that TRN CRH inputs originate from non-neuroendocrine central CRH sources. **o** Summary diagram of confirmed CRH projections to the TRN, illustrating a complex and spatially organized network of CRH-releasing afferents. Schematic brain drawings are modified from ref. 89 under a Creative Commons CC BY license. Independent observable for retrograde tracing: $n = 14$ (**a, m, n**); for anterograde tracing: BLA, $n = 4$ (**f, i**); ZID, $n = 3$ (**g, j**); MGM and SGN, $n = 4$ (**h, k**). All scale bars are 200 μm. Source data are provided as a Source data file.

retrograde Cre-dependent AAV-tdTomato (rgAAV2-FLEX-tdTomato) into the sensory TRN (Fig. 2a), we found consistent labeling in cell bodies located in the BLA (Fig. 2b), dorsal zona incerta (ZID) (Fig. 2c), and medial (MGM) and superior geniculate nucleus (SGN) (Fig. 2d). To confirm these projections, we separately injected an anterograde Cre-dependent AAV-GFP (AAV1-pCAG-FLEX-EGFP) (Fig. 2e) into the BLA, ZID, MGM, and SGN (Fig. 2f–h) and traced GFP-labeled axons back to

the TRN (Fig. 2i–k). Interestingly, these axons did not appear to terminate in the TRN, suggesting that they are collaterals of axons that continue toward additional brain regions. Furthermore, our retrograde tracing revealed additional potential sources of CRH input, such as the ventral lateral geniculate nucleus (VLGPC) (Fig. 2m), which remains to be confirmed. Notably, we did not find CRH-releasing afferents from the paraventricular region of the hypothalamus

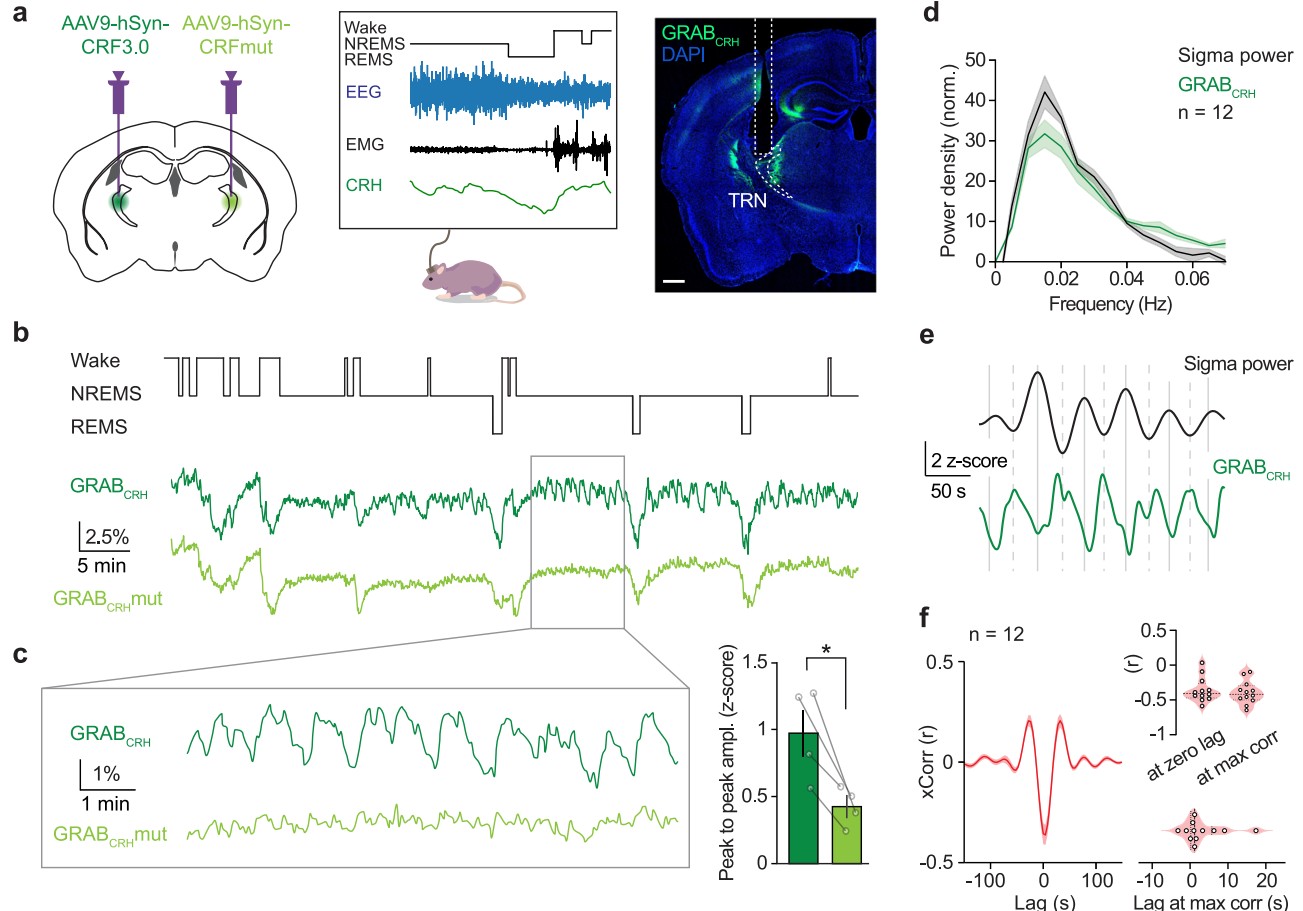

**Fig. 3 | CRH release in the TRN exhibits an infraslow oscillatory pattern during NREMS. a** Left, diagram showing stereotactic injections of the GRAB$_{CRH}$ sensor targeting TRN in one hemisphere and the CRH-insensitive control sensor, GRAB$_{CRH}$mut, in the contralateral hemisphere. Center, time-synchronized photometry recordings were carried out together with EEG/EMG recordings. Right, example image of optical fiber positioning over TRN and viral expression of GRAB$_{CRH}$ (green) (the scale bar is 500 μm). **b** Example hypnogram throughout the vigilance states with concomitant GRAB$_{CRH}$ (dark green) and GRAB$_{CRH}$mut (light green) $\Delta F/F$ signals in the TRN. Both GRAB$_{CRH}$ and GRAB$_{CRH}$mut show a decrease in relative fluorescence during wakefulness and REMS. However, during NREMS, GRAB$_{CRH}$ shows large oscillations that are not present with GRAB$_{CRH}$mut. **c** Right, close-up of GRAB$_{CRH}$ and GRAB$_{CRH}$mut during NREMS. Left, quantification of the peak-to-peak amplitude between GRAB$_{CRH}$ and GRAB$_{CRH}$mut, showing significantly different amplitudes ($n = 4$). **d** Power spectral density analysis of CRH release during NREMS shows an overlapping peak with the infraslow oscillation of sigma centered around 0.02 Hz. **e** Example trace of the time-matched CRH signal and the infraslow oscillation of sigma during NREMS. Continuous vertical gray lines indicate sigma peaks, and dotted vertical gray lines indicate sigma valleys. **f** Left, cross-correlation of the two signals shows that they anti-correlate, with maximal CRH release during NREMS overlapping with periods of low spindle activity corresponding to sleep fragility ($n = 12$). Right, the upper violin plots show the individual variability of the cross-correlation, each dot is the correlation coefficient ($r$) at lag 0 (left) and at the maximum correlation (max corr, right) for each mouse; the lower violin plot shows the distribution of the lag values at maximum correlation (lag at max corr). Data are represented as mean ± SEM or KDE for violin plots, with dashed lines indicating the median and dotted lines indicating the upper and lower quartiles. Statistical analysis was performed by a two-tailed paired $t$-test, with *$p < 0.05$, **$p < 0.01$, ***$p < 0.001$. For additional statistical information, see Supplementary Table 1. Source data are provided as a Source data file.

(paraventricular nucleus (PVN)) (Fig. 2n), the main trigger of the stress response within the hypothalamic-pituitary-adrenal axis.

To further support that CRH is released in the TRN, we used CRH-IRES-Cre × Ai27D mice, in which the TdTomato fluorescent reporter for ChR2 labels all CRH-positive neurons. We performed immunostaining for Chromogranin A, a marker of large dense-core vesicles previously used to show CRH release in the PVN[39]. Quantification of Chromogranin A puncta within TdTomato-labeled fibers revealed comparable expression levels across the TRN, BLA, and PVN (Supplementary Fig. 1).

Overall, our findings indicate that the sensory TRN receives CRH afferents from thalamic nuclei and limbic-related regions with a distinct region/sector specificity in their innervation patterns (Fig. 2o). These regions are part of the neuromodulatory central CRH system, which plays complex roles in stress, anxiety[15,40] and sleep regulation[41–44]. This suggests that the CRH input to TRN may be involved in integrating sensory and limbic information, potentially modulating TRN-related sleep rhythmogenesis.

## CRH release at the TRN exhibits an infraslow oscillatory pattern throughout NREMS

To investigate endogenous CRH dynamics in the TRN across the sleep-wake cycle, we combined fiber photometry with polysomnographic (EEG/EMG) recordings in male mice, using the CRH sensor GRAB$_{CRH}$[34]. GRAB$_{CRH}$ was virally expressed in the sensory TRN, and, to confirm its specificity for CRH binding, in a subset of recordings, a CRH-insensitive mutant version of the sensor (GRAB$_{CRH}$mut) was expressed in the contralateral TRN (Fig. 3a). In ex vivo experiments, we verified that CRH perfusion induced an increase in the fluorescence of GRAB$_{CRH}$ but not GRAB$_{CRH}$mut (Supplementary Fig. 2). A representative recording from an implanted mouse is shown in Fig. 3b, where the hypnogram

that depicts vigilance states extracted from the EEG/EMG recording is aligned with the GRAB$_{CRH}$ and GRAB$_{CRH}$mut signals acquired simultaneously. Both signals showed a similar decrease during wakefulness and REMS, as previously observed with other GRAB sensors and their mutant variants as well[4,45]. This indicates that changes in fluorescence during wakefulness and REMS were largely influenced by factors independent of CRH release, such as pH and hemodynamic changes[33,46,47]. However, during NREMS, GRAB$_{CRH}$ exhibited a large CRH-dependent fluctuation that was not present with GRAB$_{CRH}$mut (Fig. 3c). Notably, the oscillatory release of CRH during NREMS showed a periodicity of ~50 s (Fig. 3d) reminiscent of the ISO of sigma power[17]. To understand the temporal relationship between the two signals, we cross-correlated CRH release and the ISO of sigma during consolidated NREMS bouts. This revealed that CRH release and sigma power were anti-correlated (Fig. 3e, f), with the highest CRH peaks coinciding with periods of NREMS fragility.

Together, these data show that CRH release in the TRN during NREMS follows an oscillatory pattern with a 50-s infraslow periodicity that anti-correlates with sleep spindle dynamics, similar to noradrenaline[31,32]. This suggests that CRH may contribute to the neuromodulation of sleep spindles, potentially influencing the probability of interruptions during NREMS fragility periods, such as MAs.

## Optogenetic manipulation of CRH release in the TRN bidirectionally modulates NREMS fragmentation by MAs

Next, we tested whether CRH release in the TRN could influence NREMS fragmentation and the sigma power band by manipulating CRH release selectively during NREMS using optogenetic methods coupled with polysomnographic recordings.

To increase the CRH release, we used CRH-IRES-Cre × Ai27D mice. In mice implanted with EEG/EMG electrodes and bilateral optic fibers above the sensory TRN, the CRH-releasing fibers were selectively activated during NREMS. We employed a closed-loop stimulation paradigm that triggered LEDs at 465 nm to flash trains of blue light at 10 Hz for 50 s based on slow oscillations and sigma power thresholds for NREMS, adjusted for each mouse individually, followed by a pause of at least another 50 s until the light could be triggered again. The purpose of this stimulation paradigm was to entrain the endogenous oscillation of CRH release during NREMS (Fig. 4a). Figure 4b shows a representative hypnogram and the corresponding sigma and delta power for a portion of a baseline recording compared to a recording with optogenetic stimulation in the same mouse. Representative NREMS and REMS power spectra from baseline and stimulation sessions are shown in Supplementary Fig. 3a, b.

MAs were scored during NREMS when brief periods (≤12 s) of cortical activation (shift of EEG to higher frequencies without spindles) were accompanied by an increase in EMG activity[11,48]. In baseline recordings, MAs spontaneously occurred in portions of NREMS preceded by wakefulness bouts and showed a decrease in frequency as NREMS progressed toward REMS. They also coincided, as expected, with low sigma power, occurring within the periods of NREMS fragility, described by the descent of the ISO of sigma[17,26]. Photostimulation of CRH-releasing fibers significantly increased the occurrence of MAs per hour of NREMS (Fig. 4c), leading to NREMS fragmentation (Supplementary Fig. 3c–e). It further induced an overall decrease in sigma power throughout NREMS (Fig. 4d), without affecting the periodicity of sigma ISOs (Supplementary Fig. 3f, g). Delta power remained unaffected (Fig. 4e). Next, we examined intermediate sleep preceding transitions to REMS[17,49]. While the number of transitions was not affected by photostimulation (Supplementary Fig. 3h), the sigma surge preceding REMS[50,51] showed a significant decrease in the amplitude of the sigma peak (Fig. 4f, g), consistent with the decrease in sigma power throughout NREMS, and a delay in REMS entry from the sigma peak (Fig. 4h). Importantly, these effects were not observed when photostimulation was applied to wild-type mice (Supplementary Fig. 4).

These data suggest that boosting CRH release in the TRN reduces sleep spindle density and induces fragmented sleep, as indicated by increased NREMS interruptions by MAs.

To inhibit CRH release, we used a retrograde Cre-dependent virus to express the inhibitory G$_{i/o}$-coupled bistable GPCR Parapinopsin (PPO)[52] in the sensory TRN of CRH-IRES-Cre mice. PPO was activated by blue light (456 nm) at 10 Hz[52], allowing us to use the same stimulation paradigm to dampen CRH release. Mice implanted with bilateral optic fibers over the TRN and EEG/EMG electrodes showed expected viral expression of PPO-Venus. Ex vivo histology showed labeled axons in the TRN and retrogradely labeled somata in the corresponding CRH-projecting regions identified in our tracing studies (Fig. 5a, b).

Selective dampening of CRH release during NREMS reduced the occurrence of MAs (Fig. 5c, d), indicating fewer NREMS interruptions. No changes were observed in the sigma power band throughout NREMS (Fig. 5e), at NREMS-REMS transitions (Fig. 5g–i), or in the periodicity of the ISO of sigma with photoinhibition (Supplementary Fig. 5f, g). However, the delta band power increased significantly throughout NREMS (Fig. 5f). This was accompanied by reduced sleep fragmentation, as indicated by the decrease in the number of NREMS bouts (Supplementary Fig. 5d), the prolongation in NREMS bout length (Supplementary Fig. 5e), and the decrease in the NREMS-REMS transitions (Supplementary Fig. 5h). Altogether, the data indicate that reduced CRH levels in TRN consolidate and improve the depth of NREMS. Performing the same stimulation paradigm in mice injected with a yellow fluorescent control virus showed no effects (Supplementary Fig. 6), confirming the specificity of the manipulation.

Overall, these data indicate that the endogenous ISO of CRH release in the TRN, which coincides with periods of NREMS fragility, influences the occurrence of MAs within these permissive windows, contributing to NREMS fragmentation. Entraining or dampening the CRH oscillation leads to a bidirectional modulation of fragmentation and consequently of the quality of NREMS, as reflected in changes in the sigma and delta power bands and the architecture of NREMS.

## CRHR1 activation decreases low-threshold bursting in TRN neurons

To gain insight into the cellular mechanisms through which CRH modulates MAs and NREMS oscillations in the TRN, we performed ex vivo patch-clamp experiments. TRN neurons possess a specialized assembly of ion channels, synaptic receptors, and intracellular Ca$^{2+}$ mechanisms that sustain coordinated rhythmogenic bursting essential for their pacemaking properties in NREMS oscillations within the thalamocortical loop[20,21,53]. The primary ionic mechanisms underlying cyclical TRN bursting, particularly spindle generation, include low-voltage gated Ca$_V$3 Ca$^{2+}$ channels[22,54] and small-conductance Ca$^{2+}$-activated type-2 K$^+$ (SK) channels[55], both of which can be modulated by CRH[56,57]. In current-clamp whole-cell recordings from mice of either sex, injections of depolarizing current steps from a membrane potential of −70 mV elicited an oscillatory low-threshold bursting followed by sustained tonic firing (Fig. 6a). Bath application of CRH decreased the number of bursts across a wide range of depolarizing currents in a concentration-dependent manner (Fig. 6b). Although we present data by grouping recordings from either males or females, we specifically verified that the effect of CRH on TRN bursting was independent of sex (Supplementary Fig. 7a). Similar to bath-applied CRH, endogenous CRH release induced by photoactivation of CRH-positive axons in the CRH-IRES-Cre × Ai27D mouse line with a train of blue light (470 nm) at 20 Hz replicated the effects observed with exogenous CRH application. In slices from wild-type mice, the train stimulation had no effect on the bursting capacity of TRN neurons (Fig. 6c). The CRH-mediated reduction in bursting was abolished by the selective CRHR1 antagonist NBI35965 (3 μM) under both bath-applied and endogenous release conditions (Fig. 6d). These results indicate that CRH, through activation of CRHR1, can impact the firing mode of TRN neurons, a

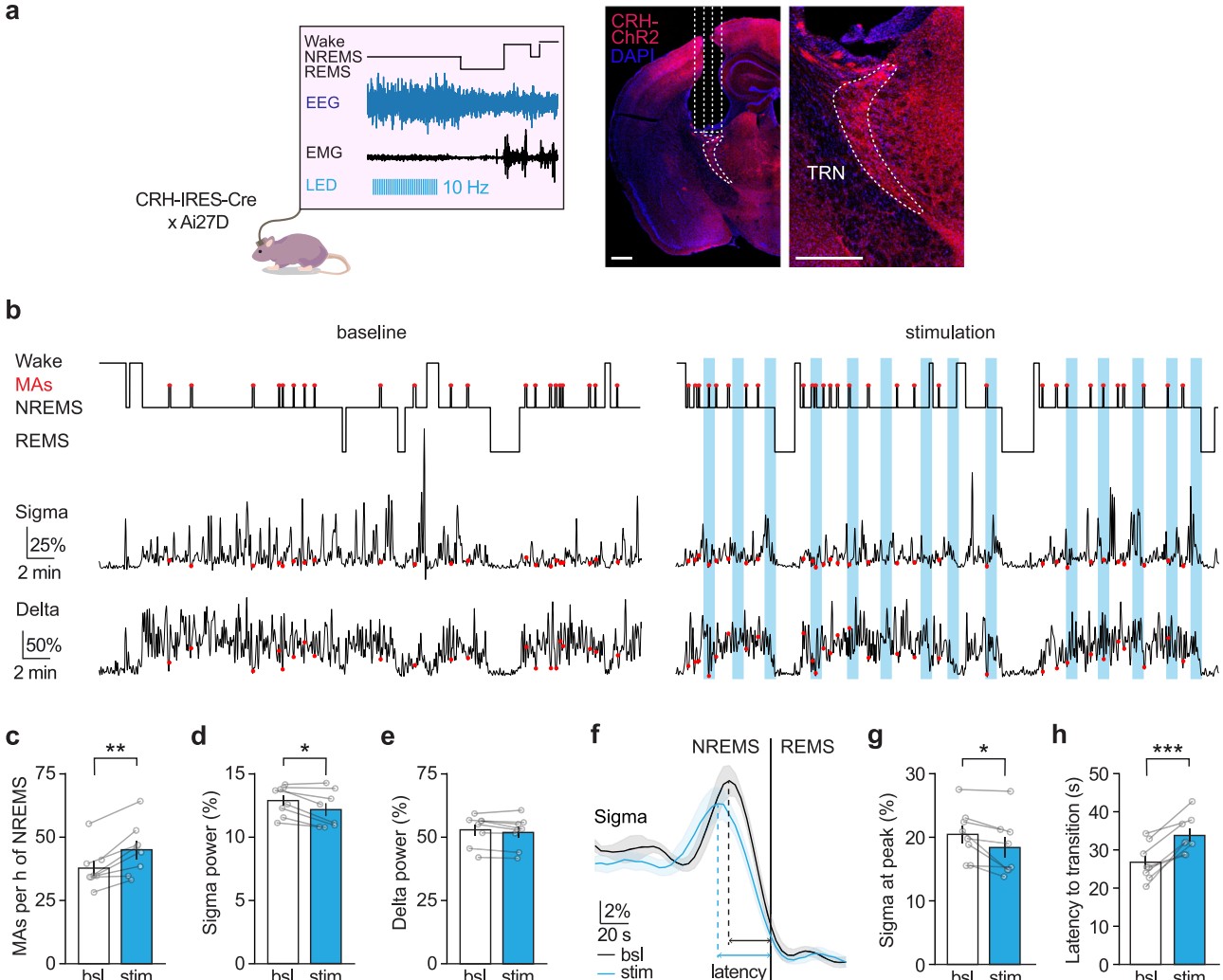

**Fig. 4 | Boosting CRH release in TRN promotes microarousals and decreases sigma power during NREMS. a** Schematic of the experimental approach for optogenetic activation of CRH release. EEG/EMG recordings were performed in ChannelRhodopsin2 (ChR2)-expressing CRH-IRES-Cre × Ai27D mice implanted with bilateral optic fibers positioned above the sensory TRN. CRH release was selectively photoactivated during NREMS using 456 nm light stimulation at 10 Hz, triggered every 50 s of closed-loop detected NREMS with a minimum interval of 50 s between stimulations. The depiction of LED train stimulation is illustrative. Representative images on the right demonstrate examples of optic fiber positioning (scale bars represent 500 μm) and expression of ChR2 in CRH-positive axons (CRH-ChR2-red). **b** Example of a polysomnographic recording with photoactivation of CRH release in the TRN during baseline and stimulation in the same mouse. The hypnograms at the top indicate the vigilance states and the occurrence of microarousals (MAs, marked by red dots), with the corresponding sigma power at the bottom. Blue-shaded areas indicate the timing of photostimulation. **c** Photoactivation of CRH release increased the number of MAs and decreased the average sigma power throughout NREMS (**d**) without change in delta power (**e**). **f** The surge in sigma power at transitions to REMS was also decreased by photoactivation of CRH release (**g**), which was accompanied by a prolonged latency to REMS from the peak of sigma (**h**). Data are represented as mean ± SEM, and $n = 8$. Statistical analysis was performed by a two-tailed paired $t$-test, with $*p < 0.05$, $**p < 0.01$, $***p < 0.001$. For additional statistical information, see Supplementary Table 1. Source data are provided as a Source data file.

mechanism associated with changes in the sigma and delta power bands[36,58].

Next, we investigated the ionic mechanisms underlying CRH-induced burst reduction. In current-clamp experiments, bath application of CRH did not alter the membrane potential (Supplementary Fig. 7b), consistent with the reported modulatory actions of CRHR1[14], indicating that the shift in burst activation was not due to neuronal hyperpolarization or depolarization. Next, we conducted voltage-clamp experiments to analyze the primary ionic currents sustaining cyclic bursting—namely, $Ca_V3$-mediated $Ca^{2+}$ currents and SK2-mediated currents[22,54,55]. In TRN cells patched with a $K^+$-methyl sulfate-based intracellular solution in the presence of the $Na^+$ channel blocker tetrodotoxin, we examined biphasic currents elicited by 40-mV depolarizations from a holding potential of −75 mV. These consisted of an inward $Ca_V3$-mediated $Ca^{2+}$ component that was curtailed by an outward SK-

mediated current (Fig. 6e). To obtain an initial assessment of the effect of CRH on these currents, we measured the charge transfer associated with the biphasic current. Since the negative phase primarily reflects $Ca_V3$-mediated depolarizing $Ca^{2+}$ influx, a reduction in $Ca_V3$ channel activity would be expected to decrease this component. Conversely, SK2 channel modulation would primarily affect the positive phase, representing repolarizing $K^+$ efflux[55]. Bath application of CRH induced a trend toward a reduction in the negative charge component ($p = 0.076$) without affecting the positive charge (Fig. 6e). Control recordings without CRH application indicate no change in either component. The CRH-induced reduction in the negative charge suggests that CRH may affect the $Ca_V3$ contribution, although the extent of the effect could be masked by the tight coupling between $Ca_V3$ and SK2 currents. To further dissect the individual contributions of these conductances, we modified our recording conditions. To isolate SK2 currents from $Ca_V3$-mediated $Ca^{2+}$

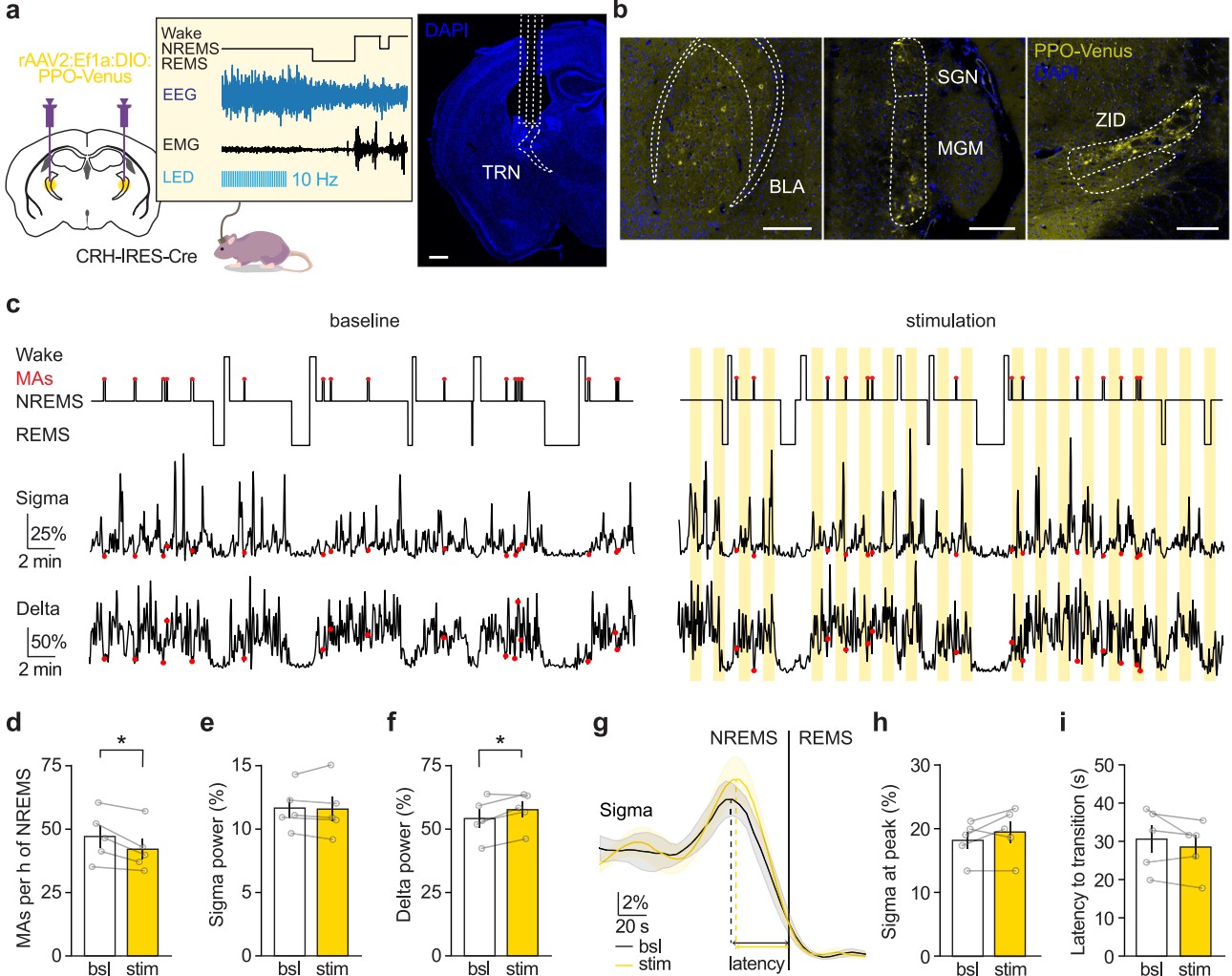

**Fig. 5 | Inhibition of CRH release in TRN consolidates NREMS. a** Schematic of the experimental approach for optogenetic inhibition of CRH release. The inhibitory opsin Parapinopsin (PPO) was virally expressed in CRH-IRES-Cre mice that targeted all CRH projections to the sensory TRN. Mice were implanted with bilateral optic fibers above the TRN and electrodes for EEG/EMG recordings. CRH release was selectively photoinhibited during NREMS using 456 nm light stimulation at 10 Hz, triggered every 50 s of closed-loop detected NREMS with a minimum interval of 50 s between stimulations. The depiction of LED train stimulation is illustrative. Representative images on the right show an example of optic fiber positioning (scale bar is 500 μm). **b** Example images from the same mouse showing the retrograde expression of PPO-Venus (yellow) in somas of the expected input regions containing CRH-releasing neurons (scale bars are 200 μm). Comparable expression was found in $n = 5$ infected mice. **c** Example of a polysomnographic recording with photoinhibition of CRH release in TRN during baseline and stimulation in the same mouse. The hypnograms at the top indicate the vigilance states and the occurrence of microarousals (MAs, marked by red dots), with the corresponding sigma power at the bottom. Yellow-shaded areas indicate the timing of photostimulation. **d** PPO decreased the number of MAs, with no effect on the average sigma power (**e**) but an increase in delta power throughout NREMS (**f**). **g** The surge in sigma power at the transition to REMS was not affected by PPO (**h**) or the latency to REMS from the sigma peak (**i**). Data are represented as mean ± SEM and $n = 5$. Statistical analysis was performed by a two-tailed paired $t$-test, with $*p < 0.05$, $**p < 0.01$, $***p < 0.001$. For additional statistical information, see Supplementary Table 1. Source data are provided as a Source data file.

influx, we leveraged the fact that SK2 currents can also be triggered by $Ca^{2+}$ influx through high-voltage-gated $Ca^{2+}$ channels. In TRN cells held at −50 mV, brief depolarizations to +20 mV elicited a tail current (Fig. 6f), which is mediated by SK2 channels[54,55]. Neither the charge nor the decay time of this tail current was affected by bath application of CRH (Fig. 6f), suggesting no significant impact of CRH on SK2 currents. To isolate $Ca_V3$ currents, we switched to a $Cs^+$-gluconate-based intracellular solution to block all $K^+$ channels, including SK2. Under these conditions, low-voltage-activated $Ca_V3$ currents were significantly reduced by CRH, an effect that was prevented by the CRHR1 antagonist NBI35965 (Fig. 6g). CRH decreased the peak amplitude of $Ca_V3$ currents by ~10%, raising the question of whether this modest effect was sufficient to explain the observed reduction in cyclic TRN cell bursting. To test this, we elicited repetitive $Ca_V3$ currents by applying three consecutive brief depolarizations (Fig. 6h). Notably, CRH had a more pronounced effect on $Ca_V3$

currents after the initial peak, consistent with its reported influence on $Ca_V3$ channel recovery from inactivation[56]. This finding on progressive attenuation of $Ca_V3$ currents aligns with the dampening of repetitive bursting observed in TRN cells upon CRHR1 activation.

Overall, the CRHR1-dependent decrease in TRN neuronal bursting through a reduction in $Ca_V3$-mediated $Ca^{2+}$ currents provides a plausible mechanism for the decrease in sigma power and the increased vulnerability to MAs observed with photoactivation of CRH afferents in vivo. These findings suggest a cellular basis for the effects of CRH on NREMS continuity, pointing to its potential role in sleep fragmentation.

**CRH modulates NREMS through activation of CRHR1 in PV+ TRN neurons**

Given the diverse sources of CRH-releasing fibers, we tested whether CRH could be co-released with glutamate or GABA in the TRN using

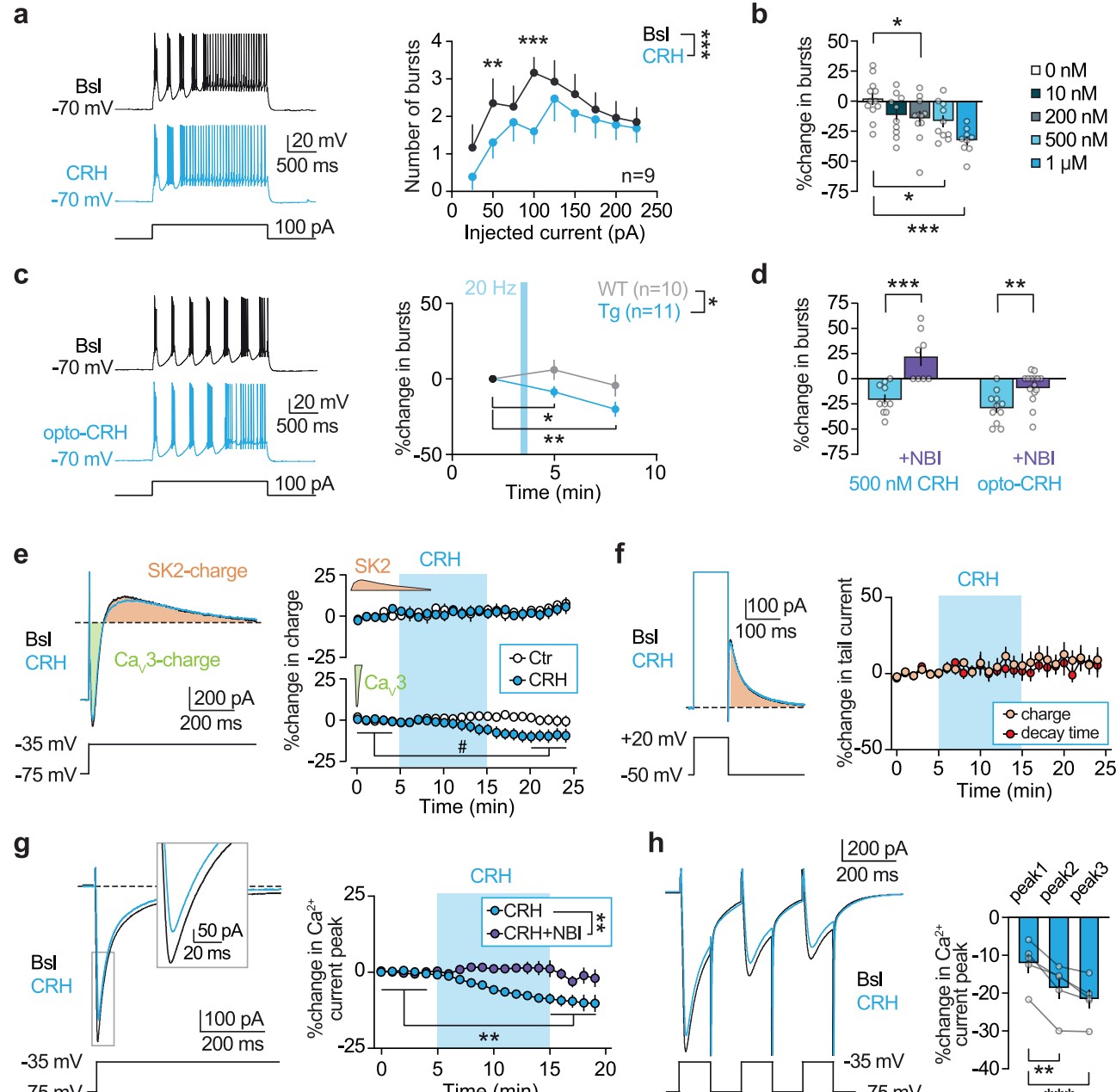

**Fig. 6 | CRHR1 activation decreases TRN cell bursting. a** Left, current-clamp recordings from a TRN neuron showing low-threshold bursts followed by tonic firing in response to depolarizing steps, before (Bsl) and after CRH application (1 μM). Right, bath-applied CRH significantly reduced bursting across a range of current steps (*n* = 9). **b** CRH decreases bursting in a concentration-dependent manner (average of 50–125 pA steps: 0 nM, *n* = 12; 10 nM, *n* = 10; 200 nM, *n* = 11; 500 nM, *n* = 10; 1 μM, *n* = 9). **c** Photostimulation of CRH afferents (TG, 20 Hz, 5-ms pulses, 22 s) mimicked the effect of exogenous CRH, with no effect in wild-type mice (WT, *n* = 10; TG, *n* = 11). **d** CRHR1 antagonist NBI35965 (NBI, 3 μM) abolished the effects of CRH (CRH, *n* = 10; NBI + CRH, *n* = 8; Opto-CRH, *n* = 11; NBI + Opto-CRH, n = 15). **e** Left, example biphasic currents in a TRN neuron before and after CRH. Ca$_V$3-channel- and SK2-channel-mediated contributions to total charge are color-coded. Right, CRH only affected the negative component (*p* = 0.076), otherwise stable in control recordings (Ctr, *n* = 7; CRH, *n* = 8). **f** Left, example tail currents mediated by SK2 channels at the end of depolarizations to +20 mV. Right, percentage change of charge and decay time, indicating no significant impact of CRH on SK2-mediated currents (*n* = 7, *p* > 0.05). **g** Left, isolated Ca$_V$3 currents (magnified portion in the inset). Right, CRH (500 nM, *n* = 9) induced a decrease in Ca$_V$3 currents that was prevented by NBI35965 (3 μM, *n* = 6). **h** Repetitive depolarizations revealed a progressive decrease in Ca$_V$3 current peaks following CRH application (*n* = 5). Data are mean ± SEM. Voltage-clamp traces are the average of 12–15 recordings. Statistical tests: **a**, **c**, **e**, **g** repeated-measures two-way ANOVA, **b** one-way ANOVA, and **h** repeated-measures one-way ANOVA with Holm–Šídák's test; **d** two-tailed Mann–Whitney test, **e** two-tailed paired *t*-test, and **f**, **g** two-tailed Wilcoxon test. #*p* < 0.1, *\*p* < 0.05, *\*\*p* < 0.01, *\*\*\*p* < 0.001. See Supplementary Table 1 and Source data file for details.

ex vivo recordings. In slices of CRH-IRES-Cre × Ai27D mice, we monitored the occurrence of synaptic responses during photostimulation of ChR2 in CRH-positive axons that contact the TRN (Supplementary Fig. 8). Recordings conducted using a CsCl-based intracellular solution in the presence of ionotropic glutamate receptor antagonists—

designed to isolate GABAergic inputs—revealed that photostimulation failed to evoke inhibitory postsynaptic currents (IPSCs; Supplementary Fig. 8a). In contrast, recordings performed with a standard KGluconate-based solution in the presence of a GABA$_A$ receptor blocker showed that ~50% of TRN neurons exhibited excitatory

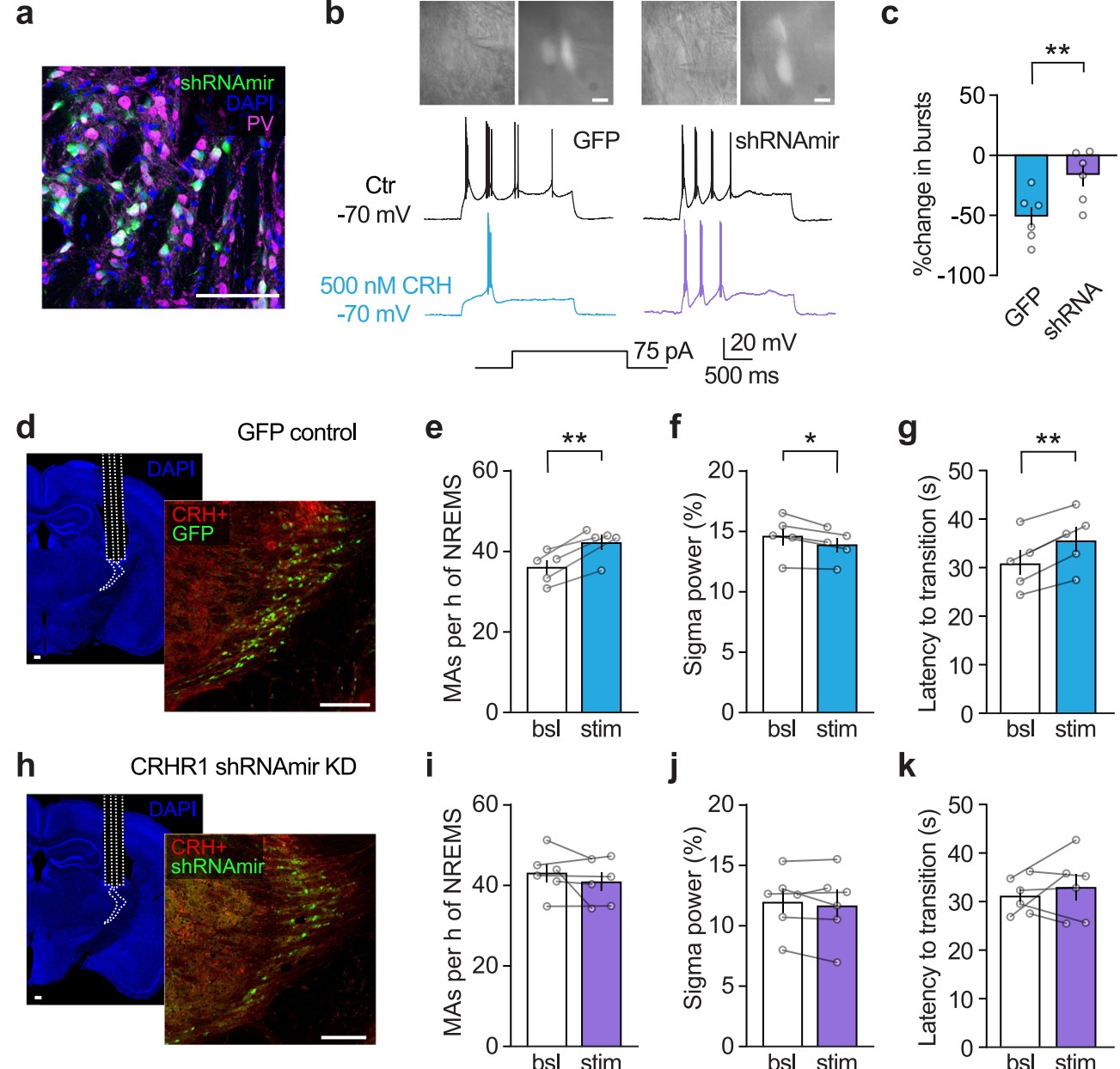

**Fig. 7 | CRH modulates NREMS through activation of CRHR1 in PV+ TRN neurons. a** Representative image showing the expression of the shRNAmir virus used to knock down (KD) CRHR1 in TRN. The GFP reporter for shRNAmir expression (green) is colocalized with parvalbumin (PV−magenta) in TRN neurons (scale bars are 200 µm). Comparable expression was found in n = 6 infected mice. **b** Example images (infrared and fluorescence channel) of patched-clamped TRN neurons from mice injected with a control GFP or shRNAmir (scale bars are 10 µm). Example traces from current-clamp recordings showing that CRHR1 KD prevented the changes in TRN bursting induced by bath application of CRH (500 nM).
**c** Quantification of the CRH-induced decrease in TRN bursting, which was abolished by the CRHR1 KD. **d** Example images for the GFP injected controls, left−optic fiber implantation over the sensory TRN in CRH-IRES-Cre × Ai27D mice implanted for EEG/EMG recordings and right−channelrhodopsin2 (ChR2−red) expression in CRH-positive axons and GFP (green) labeled TRN neurons. Scale bars are 200 µm. **e** In control mice injected with GFP, photoactivation of CRH release during NREMS increased the number of MAs, decreased the average sigma power (**f**), and prolonged the latency to REMS from the sigma peak (**g**). **h** Example images for the shRNAmir injected mice, left−optic fiber implantation over the sensory TRN in CRH-IRES-Cre × Ai27D mice implanted for EEG/EMG recordings and right−ChR2 (red) expression in CRH-positive axons and the shRNAmir expression labeled with GFP (green) in TRN PV+ neurons. Scale bars are 200 µm. **i**−**k** CRHR1 KD abolished the NREMS fragmentation induced by photoactivation of CRH release. Data are represented as mean ± SEM. Statistical analysis was performed by (**b**) one-tailed unpaired *t*-test and (**e**−**g**, **i**−**k**) one-tailed paired *t*-test with *$p < 0.05$, **$p < 0.01$, ***$p < 0.001$. For additional statistical information, see Supplementary Table 1. Source data are provided as a Source data file.

postsynaptic potentials (EPSPs) that were time-locked to flashes of blue light (Supplementary Fig. 8b), indicating that some CRH inputs to the TRN co-release glutamate. However, a comparison of these responses with EPSPs elicited by photostimulation of layer six inputs in Ntsr1-Cre × Ai27D mice (see "Methods") revealed striking differences. Glutamatergic inputs photoactivated in CRH-IRES-Cre × Ai27D mice were of small amplitude and did not induce low-threshold bursting, even with repeated LED flashes of 1−5 ms. In contrast, cortical inputs evoked large-amplitude EPSPs that, with LED flashes as brief as 100 µs, reliably summated to trigger TRN bursting (Supplementary Fig. 8c).

Although further characterization of the chemical identity of CRH synapses onto TRN in an input-specific manner is warranted, these

initial findings suggest that the glutamatergic input provided by the CRH-releasing fibers is unlikely to impact TRN burst propensity during NREMS. However, the question remains as to whether the effects of photostimulation observed in vivo are strictly dependent on CRH release or whether other neurotransmitters that are co-released contribute.

To directly assess the role of CRHR1 in the modulation of NREMS through the TRN, we knocked down (KD) CRHR1 specifically in PV+ TRN neurons using a virally delivered short hairpin RNA embedded in a microRNA backbone (shRNAmir). The virally encoded shRNAmir (AAV9-S5E2p-EGFPm-CRHR1-shRNAmir) was designed to express only under the PV promoter. This approach achieves TRN-specific expression while targeting the main spindle pacemaking TRN cellular population[37], which also exhibits the highest expression of CRHR1. The effectiveness of the KD was validated functionally in ex vivo electrophysiological recordings, given the lack of specificity of CRHR1 antibodies[59]. Figure 7a shows a representative image of the viral expression of shRNAmir bound to GFP confined to PV+ neurons. Ex vivo recordings were carried out in slices from mice, >5 weeks after infection with a control GFP or the shRNAmir virus to ensure sufficient degradation of CRHR1. Bath application of CRH reduced bursting in control TRN neurons, whereas this effect was prevented in slices from mice injected with the shRNAmir virus (Fig. 7b, c). These results indicate that the shRNAmir virus has a consistent expression with the expected activity under the PV promoter and that mRNA interference and subsequent KD of CRHR1 from the plasma membrane were sufficient to prevent CRH from influencing TRN neuron intrinsic excitability.

Next, we tested the impact of CRHR1 KD on CRH-mediated effects during NREMS using optogenetic stimulation in CRH-IRES-Cre × Ai27D mice injected with either a control GFP virus or the shRNAmir virus bilaterally in the sensory TRN during EEG/EMG recordings. CRHR1 KD did not affect sleep architecture under baseline conditions (Supplementary Fig. 9a–c). However, shRNA-infected mice displayed lower sigma power compared to control mice (Supplementary Fig. 9d), without changes in delta power (Supplementary Fig. 9e), suggesting that chronic CRHR1 KD may induce compensatory alterations in TRN rhythmogenesis. Next, we tested the effects of acute CRH boosting during NREMS. Both groups received the same stimulation paradigm used in the ChR2 experiment to selectively increase CRH release in TRN during NREMS. Mice injected with the GFP control virus responded to the photoactivation of CRH fibers with an increase in the number of MAs, decreased sigma power throughout NREMS, and longer latency to REMS entry (Fig. 7d–g), consistent with previous observations and confirming that the GFP control virus has no impact on the parameters quantified in these recordings. In contrast, mice with CRHR1 KD in PV+ TRN neurons (Fig. 7h) showed no changes in MAs, the sigma power band, or the latency to REMS entry following photostimulation (Fig. 7i–k). Together, these data indicate that CRHR1 in PV+ TRN neurons is causally involved in CRH-mediated modulation of NREMS fragmentation and changes in correlates of spindle density.

## Discussion

We describe CRH circuits originating in thalamic and limbic-related regions that interact with the TRN to prime the occurrence of MAs during NREMS, thereby modulating NREMS fragmentation. Sleep fragmentation is a commonly occurring sleep disturbance, comorbid with various psychiatric and non-psychiatric conditions[2,18], with highly detrimental consequences on cognitive processing, memory, and mood regulation[18,29,30]. It has also been closely linked to stress and anxiety[2]. CRH, the main stress- and anxiety-related peptide, is released from numerous brain regions across the central nervous system[14,15], and has long been hypothesized to play a role in sleep regulation[60]. Our study describes how NREMS can be modulated by CRH through its action on the TRN, a key component of the thalamocortical loop,

which serves as the main NREMS pattern generator. These findings contribute to our understanding of the neuromodulatory mechanisms that govern the continuity of NREMS, where CRH may act as an integrator of anxiety through its interaction with the thalamocortical loop.

A few recent studies have begun to uncover the heterogeneous effects of central CRH circuits in modulating the sleep-wake cycle, including CRH neurons in the preoptic area that promote NREMS[44] and those in the dorsomedial medulla that promote REMS[41]. Moreover, CRH neurons in the subthalamic nucleus[42] and nucleus accumbens[43] have been involved in REMS regulation, particularly in response to stress. However, despite evidence linking CRH to stress integration and sleep fragmentation, surprisingly little is known about its direct role in modulating NREMS[2,14].

Endogenous CRH release in the TRN during NREMS fluctuates with a periodicity of 50 s (0.02 Hz), anti-correlating with sigma power. The ISO of sigma power is a reliable marker for distinguishing the two alternating NREMS substates of continuity, where memory consolidation can occur, and fragility, with an increased probability of NREMS interruptions such as MAs or transitions to other vigilance states[17]. Recent studies have reported infraslow activity patterns during NREMS in physiological markers of arousal[26,61,62], hemodynamic parameters[63], and neuromodulators, including orexin[64,65], noradrenaline[4,31,32] and serotonin[66]. Moreover, neuronal activity on the infraslow scale has been observed in the glutamatergic neurons of the preoptic area of the hypothalamus[67] and the GABAergic neurons of the dorsomedial medulla[68].

Increased CRH release in the TRN during NREMS fragility periods coincides with an increase in autonomic markers of arousal, such as heart rate[26] and pupil diameter[62], reflecting the higher arousability and predisposition to MAs in this substate[17]. The oscillatory release pattern of CRH and its temporal alignment with sigma power at the TRN parallels the behavior of noradrenaline, where both modulators limit TRN bursting[69] and impact NREMS continuity by decreasing sigma power and increasing MAs[4,32]. These findings suggest that CRH release dynamics in the TRN and the subsequent effect on MAs are consistent with the infraslow rhythmic activity observed with other arousal-related neuromodulators, contributing to the microarchitecture of NREMS. However, the source of this rhythmic pattern remains an open question. The fluorescent changes captured by GRAB$_{CRH}$ in TRN neurons may result from a composite release pattern from multiple CRH sources, and it remains unclear whether all CRH afferents to the TRN are active during NREMS. Future investigation of the individual CRH sources and their potential to exhibit infraslow oscillatory activity could clarify their role in sleep regulation and their relationship to the sigma-band ISOs.

Photoactivation of CRH fibers induced an increase in MAs and a reduction in sigma power, consistent with a fragmented NREMS that is associated with detrimental effects on cognition and memory[18]. The sleep fragmenting effects of CRH were causal to CRHR1 activation in PV+ TRN neurons, supporting the role of CRH in modulating NREMS continuity and spindle rhythmogenesis[35–37]. At the cellular level, activation of CRHR1 decreased the number of low-threshold Ca$^{2+}$ bursts in TRN neurons, representing a plausible mechanism for the reduction in sigma power observed in vivo. Whether the decrease in bursts may also contribute to the increased vulnerability of NREMS fragility periods for MAs requires further examination, as the cellular and circuit basis of the manifestation of MAs is not clear. In contrast, photoinhibition of CRH release in TRN decreased the number of MAs and coincided with an increase in delta power, consistent with an improved NREMS consolidation[70]. These data indicate that CRHR1 activation promotes MAs, whereas reducing CRH release in the TRN enhances the depth of NREMS, potentially due to a lack of synergy between CRH and other neurotransmitters, which remains to be explored.

CRHR1 KD through shRNAmir interference abolished the ability of CRH to decrease bursts in spindle-generating PV+ TRN neurons, thus

preventing the sleep fragmentation effects of CRH photostimulation in vivo. These findings indicate that CRHR1 in PV+ TRN neurons is essential for CRH-mediated modulation of NREMS, underscoring the critical role of the CRH-CRHR1 system. The complete loss of effects on sleep fragmentation in CRHR1 knock-down mice supports the conclusion that these effects are primarily mediated by CRHR1 activation in the TRN, independent of other synaptic neurotransmitters or neuromodulators that may be co-released with CRH. Our data suggest that CRH afferents to the TRN can also release glutamate, at least onto a subset of TRN neurons. Although this excitatory input does not appear to influence TRN sleep rhythmogenesis, these findings underscore the need for a more detailed investigation into the chemical identity of the CRH-releasing inputs. Future studies should aim to characterize the endogenous activity patterns of individual CRH projection pathways during sleep and determine their complete neurochemical profile to clarify their specific roles in sleep regulation.

The experiment with CRHR1 KD in TRN addresses another potential confounding factor, that is, the non-specific activation of CRH afferents to regions other than the TRN during photostimulation. The absence of any effect in CRHR1 KD mice confirms that there are no significant off-target effects from the optogenetic stimulation, further affirming the specificity of CRHR1 activation in PV+ TRN neurons for the observed results.

Increasing CRH release with photostimulation also reduced sigma surge at the transition to REMS and delayed REMS onset, mirroring the increased REMS latency observed in stress- and anxiety-related disorders such as generalized anxiety and PTSD[1,71]. In humans, reduced spindle density at the transition to REMS disrupts the overnight amygdala adaptation during REMS, leading to daytime hyperarousal seen in various psychiatric disorders[72]. These findings suggest that CRH may contribute to 'restless REMS' and related disorders. Investigating whether anxiety-inducing events during wakefulness alter CRH dynamics during subsequent sleep could provide insights into how emotional experiences influence the neuromodulatory tone during NREMS and impact limbic adaptation during REMS.

This study provides evidence for a role of central CRH circuits in modulating NREMS fragmentation and MAs by affecting thalamocortical rhythmogenesis. CRH arousal-related neuromodulation of TRN during NREMS contributes to the fragility of NREMS, whereby increasing or decreasing the neuromodulatory tone of CRH, fragmented or consolidated NREMS. NREMS fragmentation with frequent MAs is known to have detrimental effects on cognition, memory consolidation, and mood regulation[18,73]. Our findings show that various CRH afferents contact the TRN, which may represent a hub for CRH modulation of NREMS, opening up several hypotheses for future studies. In particular, the heterogeneity of CRH circuits calls for investigations with circuit specificity to elucidate their individual roles in NREMS modulation. Considering the associations between CRH and anxiety[74–76], future studies could explore the interaction between central CRH circuits and TRN in individuals with varying levels of trait anxiety or in animal models of stress-induced anxiety. This could reveal whether CRH contributes to anxiety-related sleep impairments[2] and subsequent alterations in sleep-dependent memory consolidation[1,24,71].

Finally, while our focus was on CRH release during sleep, the effects of CRH on TRN processing during wakefulness, particularly its impact on well-known functions, such as sensory selection[77] and attentional gating[78,79], remain to be explored. Understanding these interactions will be essential for comprehensively addressing the role of CRH across vigilance states and its potential implications in anxiety disorders.

## Limitations of the study
Male and female mice were used throughout the study, except for the EEG sleep recordings, where only male mice were used. No sex differences were found in the level of CRHR1 mRNA in TRN neurons, nor in the impact of CRH on TRN bursting. However, differences could also arise in differential signaling pathways or receptor trafficking[80]. Additionally, the described CRH circuits could also present sex differences at other levels, such as in the number of CRH neurons or in their functional role[81]. To fully understand the role of CRH in the modulation of NREMS continuity in female mice through TRN, further investigation is necessary.

Additional studies are necessary to fully elucidate the identity of CRH-projecting regions to the TRN, particularly in light of the technical limitations associated with the genetic tools used in our study. Our approach, which involved Cre-dependent retrograde viral tracing or opsin expression in combination with the CRH-IRES-Cre mouse line, could potentially result in false-positive labeling due to persistent Cre expression in adulthood following transient CRH expression during early development[82]. While there is evidence supporting sustained CRH expression throughout development in regions such as the BLA[83], MGM, and ZI[82,84], it remains to be verified whether CRH expression in the SGN and VLGPC is transient or persists into adulthood.

On the other hand, our tracing approach may not have captured all potential CRH sources projecting to the TRN. Notably, our experiments primarily targeted the sensory TRN, leaving CRH projections to the anterior TRN unexamined. Moreover, the rgAAV2 viral construct used in our study has been reported to inefficiently transduce specific neuronal circuits and subpopulations, such as corticothalamic neurons and dopaminergic neurons in the substantia nigra[85]. To address these limitations, alternative viral tools with improved tropism will be necessary to identify fully the CRH-projecting inputs to the TRN.

The photostimulation protocol we designed aimed to entrain or dampen oscillatory CRH release in TRN during NREMS, but we were unable to measure CRH release with GRAB$_{CRH}$ during photostimulation due to the overlap in the detection range of the CRH sensor and the activation wavelength of the opsins. Therefore, it is unclear whether the stimulation paradigm was successful in preserving the endogenous dynamics of CRH release and the importance of CRH optogenetic manipulations during specific NREMS substates. This limitation also prevented us from validating whether photostimulation induced fluctuations in CRH release. Although both photostimulation in WT mice and CRHR1 KD prevented the effects observed with photoactivation of CRH axons, a direct measurement of CRH release induced by photoactivation in these conditions remains to be confirmed—an endeavor that may be facilitated by further development of a red-shifted neuropeptide sensor.

## Methods
### Mouse lines
Animal care and experimental procedures were carried out in accordance with the Swiss Federal Guidelines for Animal Experimentation and were approved by the Cantonal Veterinary Office Committee for Animal Experimentation (Vaud, Switzerland).

Mice of C57BL/6J genetic background were used between postnatal days P28-84, from the following transgenic lines: B6(Cg)-Crh$^{tm1(cre)}$$^{Zjh}$/J (CRH-IRES-Cre, The Jackson Laboratory, RRID:IMSR_JAX:012704), B6.Cg-Gt(ROSA)26Sor$^{tm27.1(CAG-COP4*H134R/tdTomato)Hze}$/J (Ai27D, The Jackson Laboratory, RRID:IMSR_JAX:012567). Mice of either sex were used for ex vivo experiments, whereas for in vivo experiments, only male mice were used. The CRH-IRES-Cre line expresses Cre recombinase under the CRH promoter. The Ai27D line contains a floxed ChannelRhodopsin2 (ChR2) fused to the fluorescent protein TdTomato. Homozygous CRH-IRES-Cre mice were bred with Ai27D mice to drive ChR2 and TdTomato expression in CRH-releasing neurons. Ai27D mice, regardless of their genotype, were used as wild-type controls in electrophysiology experiments. In a subgroup of experiments, B6.FVB(Cg)-Tg(Ntsr1-cre)GN220Gsat/Mmucd mice (Ntsr1-Cre, The Jackson Laboratory, RRID:MGI:4358487) were crossed with Ai27D to

drive ChR2 expression in layer 6 cortical neurons[86]. Mice were generally group-housed in 2–4 per cage, unless implanted for polysomnographic recordings, in which case they were single-housed. Food and water were provided *ad libitum*. Housing colonies had a 12-h light/dark cycle starting at 7 am, ambient temperature of 22 ± 2 °C, and humidity of 55 ± 10%. Genotyping was performed by qPCR using tissue from ear punches by Transnetyx (Tennessee, USA).

## Surgical procedures
Surgeries for viral infection, optic fiber placement, and EEG/EMG implants were performed on head-fixed mice (7–9 weeks old) in a stereotaxic apparatus under isoflurane anesthesia (2.5% for induction and 1–2% for maintenance, mixed with oxygen) and with eye-protecting gel (Viscotears). Analgesia was ensured by intraperitoneal (i.p.) injection of Buprenorphine (0.1 mg/kg) before and after surgeries, as well as local injection of a 1:1 mix of Lidocaine (6 mg/kg) and Bupivacaine (2.5 mg/kg) under the scalp before the incision and around the site of the surgery at the end. The head was fixed on a stereotactic frame, and the position was adjusted based on a minimum difference of 0.05 mm between bregma/lambda and medio-lateral coordinates. All brain region coordinates were taken as anteroposterior (AP), mediolateral (ML), and dorsoventral (DV) coordinates from bregma at the skull surface. To deliver the viral constructs, the Nanoinject III (Drummond Scientific) was used with glass capillaries back-filled with mineral oil, pulled at a DMZ puller (Zeitz-Instruments) with ~20 μm tip diameter. The injection delivery speed was set at 1 nl/s with at least a 10 min pause before retracting the pipette from the injection site. Operated mice received 1 mg/1 ml of paracetamol in drinking water during the post-surgery week.

## EEG/EMG electrodes and optic fiber implantation
For EEG/EMG experiments, head-fixed mice had two EEG screw-electrodes (Pinnacle Technology Inc.) implanted in the skull over the frontal cortex and one common screw-electrode over the occipital cortex, forming two frontal-occipital derivations for all EEG recordings. The EEG screw-electrodes were connected to a headmount (Pinnacle Technology Inc.) by soldering silver wires of the electrodes onto the headmount plate. EMG electrodes connected to the headmount were inserted under the skin above the nuchal muscles for the acquisition of muscle tone. Optic fibers for photometry recordings or cannulas for optic fibers in optogenetic manipulation experiments were implanted bilaterally over the TRN (AP: −1.1; ML: ±2.36; DV: −3.37), detailed in the corresponding sections below. All implants were cemented together and fixed to the skull with C&B Metabond (Parkell) and Vertise Flow (Kerr Dental).

## Histology
Mice were injected i.p. with 150 mg/kg of pentobarbital for intraventricular perfusion of Ringer physiological solution mixed with heparin, followed by fixation with paraformaldehyde (PFA) 4%. The brains were extracted and placed in 4% PFA for an additional 24 h of post-fixation, followed by at least 2 days in a 30% sucrose-PBS solution at −4 °C. The fixed brain samples were snap frozen in isopentane and kept at −80 °C until they were sliced with a cryostat (CM3050 S, Leica) to a thickness of 30–40 μm for tracing and EEG experiments. Slices were placed in a cryoprotectant medium and kept at −20 °C until further processing. Finally, free-floating slices were washed with PBS, labeled with DAPI for nuclei at room temperature (RT) (Invitrogen 1:5000) for 5 min, rinsed, and mounted with Fluoromount-G (SouthernBiotech). Images were taken with a Leica SP8 confocal microscope or with an Olympus VS200 Slide Scanner.

## mRNA quantification with RNAscope
Ai27D mice of 8 weeks were decapitated under deep isoflurane anesthesia, and the extracted brains were snap frozen in isopentane at −30 °C and kept at −80 °C until further processing. Coronal slices containing the TRN of 18 μm thickness were cut at the cryostat, for a total of four slices containing the TRN for each mouse, and adhered to Superfrost plus glass slides (Electron Microscopy Science). The slides were stored at −80 °C until further processing.

Fluorescent in situ hybridization of mRNA was performed using RNAscope (Advanced Cell Diagnostics) according to the manufacturer's instructions and using the solutions provided in the kits. Slides were immersed in 4% PFA for 15 min at RT, then rinsed in PBS, followed by dehydration of the samples by submersion in three ethanol solutions of increasing concentration of 50/70/100% for 5 min each. The slides were incubated with hydrogen peroxide at RT for 10 min and rinsed with nanopure water, followed by incubation with Protease IV solution for 30 min at RT, which was rinsed with PBS. The slides were incubated with the mRNA probes at 40 °C in a humidity chamber for 2 h. The probes used were the following: CRHR1-C1 (catalog nr. 418011), PV-C3 (catalog nr. 421931-C3), SOM-C4 (catalog nr. 404631-C4). After hybridization of the probes, the samples were subjected to a series of amplification steps, using Amp1, 2, and 3, for 30 min at 40 °C in the humidity chamber, except for the latter, for 15 min. Fluorescent probes (Opal 570 for C1, Opal 690 for C3, and Opal 520 for C4) were hybridized with each of the mRNA probes, followed by DAPI staining of the nuclei for 30 s at RT and immediate mounting and sealing with ProLong Gold Antifade mounting media. Negative and positive controls were performed alongside the experimental samples, processed in the same way, except for skipping the hybridization of the mRNA probes or hybridization with positive control mRNA probes as given in the RNAscope kit, respectively.

The slides were kept at 4 °C until imaging at the Leica SP8 confocal microscope, where images of the entire TRN were taken from all slices, along with BLA and CA3 for comparison of CRHR1 mRNA levels. The acquired images were processed in ImageJ Fiji, using a script in the Macro language. DAPI, PV, and SOM channels were segmented to create masks used in the quantification of the CRHR1 puncta, which were detected using the maxima with a prominence of 35 after background subtraction adjusted for the mean fluorescence −2 SD, as taken from the histogram of the fluorescence intensity distribution from each mouse. The density of CRHR1 mRNA puncta (number of puncta/total area of segmentation mask) was determined for all segmentation masks and for the intersection between different masks, normalized by area, resulting in the number of puncta/μm². The resulting CRHR1 densities from multiple images were averaged per mouse.

## Immunocytochemistry
Five brains from CRH-IRES-Cre × Ai27D mice (3 males, 2 females) were processed as described in the histology section above. Coronal brain sections (40 μm thick) corresponding to the TRN, BLA, and PVN, based on the mouse brain atlas, were used. Free-floating sections were immunolabeled for Chromogranin A, parvalbumin, and DAPI. Sections were briefly rinsed in PBS, blocked for 1 h in PBS containing 0.3% Triton X-100 (Sigma-Aldrich) and 5% normal donkey serum (Jackson ImmunoResearch), and incubated overnight at 4 °C with rabbit anti-Chromogranin A antibody (abcam, #ab15160, Lot GR3205971-6; 1:250) and mouse anti-parvalbumin antibody (Synaptic Systems, #195011, Lot 1-13; 1:500). After PBS washes, sections were incubated for 2 h at RT with the following secondary antibodies: donkey anti-rabbit IgG Alexa Fluor 647 (ThermoFisher Scientific, #A31573; 1:1000) and goat anti-mouse IgG Alexa Fluor 488 (ThermoFisher Scientific, #A11029; 1:1000). Sections were then incubated with DAPI (Sigma, 1:10000) for 10 min, rinsed, and mounted using Fluoromount-G (SouthernBiotech).

Two images per animal were acquired in the PVN, BLA, and TRN regions using a Leica SP8 confocal microscope with a 63× objective. The TRN was identified based on its characteristic parvalbumin immunofluorescence pattern.

Quantification of Chromogranin A signal intensity was performed using Fiji (ImageJ, NIH). Regions of interest (ROIs) were manually delineated based on TdTomato fluorescence, which labeled individual CRH fibers. One fiber was outlined per image, and Chromogranin A signal intensity was measured within each ROI.

## Anatomical tracing

For retrograde tracing of CRH projections to the TRN, 7–9 weeks old CRH-IRES-Cre mice were head-fixed on a stereotactic frame as described in the general surgical procedure and viral injections. To label CRH projections and mark the injection site, a 1:1 mix of rgAAV2-FLEX-tdTomato ($1.2 \times 10^{13}$ GC/ml, Addgene) and AAV2-hSyn-EGFP-WPRE ($2.8 \times 10^{13}$ GC/ml, Vector Biolab) for a total volume of 25 nl was injected unilaterally, targeting the sensory TRN (AP: −1.1; ML: −2.36; DV: −3.47).

To anatomically confirm the candidate regions from the retrograde tracing, an anterograde tracer AAV1-pCAG-FLEX-EGFP-WPRE ($1.8 \times 10^{13}$ GC/ml, Addgene) was used. The anterograde tracer was injected into the candidate regions with a volume of 100–200 nl following the same procedure as the retrograde tracing with the following coordinates: BLA (AP: −1.12; ML: −3.28; DV: −4.95), SGN and MGM (AP: −3.25; ML: −1.9; DV: −3.1), ZID (AP: −2.2; ML: −1.6; DV: −3.75).

In both experimental types of tracing experiments, mice were left for at least 2 weeks for viral expression before perfusion to collect the brain samples, which were processed as described in transcardial perfusions and histological preparations.

The acquired images were superimposed to the mouse brain atlas[87], or semi-automatically aligned to the Allen Mouse Brain Atlas, using a Fiji plug-in ABBA (developed by Bioimaging and Optics platform at EPFL)[88] to identify the brain regions expressing fluorescent markers.

## Fiber photometry and EEG/EMG recordings

For combined fiber photometry (FP) and EEG/EMG recordings, CRH-IRES-Cre male mice of 7 weeks were injected stereotactically as described in surgical procedures, with AAV9-hSyn-CRF3.0 (GRAB$_{CRH}$, $3.2 \times 10^{13}$ GC/ml, WZ Biosciences) 300 nl, bilaterally in TRN (AP: −1.1; ML: ±2.36; DV: −3.47) or with GRAB$_{CRH}$ in one hemisphere and AAV9-hSyn-CRFmut (GRAB$_{CRH}$mut, $3.2 \times 10^{13}$ GC/ml, WZ Biosciences) in the contralateral hemisphere. Mice were left to recover and express the GRAB sensors for at least 2 weeks prior to the final surgery, where EEG/EMG electrodes and optic fibers were implanted. Optic fibers (400 μm diameter, 0.66 NA, Doric Lenses) were placed above the TRN, at the same coordinates as the viral injection, but -100 μm higher on the DV axis, guided by the increase in fluorescence of the live signal from the GRAB sensors. After 1 week of recovery from surgery, the mice were placed in the recording cages, where a preamplifier (Pinnacle Technology Inc.) was connected to the headmount implant, tethering them to the recording cage. Mice were habituated in the recording cages, to the FP patch cord, and handling related to the recording procedure each day for at least 15 min per mouse for at least a week prior to the experimental recordings.

Simultaneous EEG/EMG (Sirenia acquisition system, Pinnacle Technology Inc.) and FP recordings (Doric Lenses system) were performed for up to 2 h per session during the light phase (between 8 am and 3 pm). For FP, the 465 nm channel was used to acquire the GRAB$_{CRH}$ and GRAB$_{CRH}$mut signals, set between 0.1 and 2 V. The EEG/EMG and FP recordings were time-synchronized using a TTL input generated from the FP console and fed into the EEG/EMG acquisition system. At the end of the experiments, the brains were collected as described in the surgical procedures to assess viral expression and placement of optical fibers. Data were processed in Sirenia and custom scripts in Python as described below.

## Ex vivo validation of the GRAB$_{CHR}$ sensor

After at least 3 weeks of viral infection with GRAB$_{CRH}$ and GRAB$_{CRH}$mut, mice were anesthetized with 5% isoflurane in carbogen (95% $O_2$/5%

$CO_2$) and euthanized by decapitation. Acute horizontal 250 μm-thick slices containing the TRN, as described in the ex vivo electrophysiology methods section. Live fluorescence imaging was performed on acute slices at a Zeiss LSM 710 microscope with an image acquired every 5 s. Slices were superfused with oxygenated standard artificial cerebrospinal fluid (aCSF) containing (in mM): 130 NaCl, 25.9 $NaHCO_3$, 2.48 KCl, 1.25 $NaH_2PO_4$, 1.2 $MgCl_2$, 2 $CaCl_2$, 18 glucose, and 1.7 L(+)-ascorbic acid, at RT. CRH was bath-applied after 5 min of baseline imaging. In a subset of recordings, slices were preincubated with αH-CRH. Analysis was performed by measuring mean fluorescence intensity values in ROIs in Fiji (ImageJ, NIH), expressed as %$\Delta F/F_0$, where $F_0$ is the average intensity during baseline.

## Sleep scoring

The EEG/EMG signals were acquired with a highpass filter of 0.5/10 Hz and a lowpass filter of 200/300 Hz, digitized at 2 kHz with a data conditioning and acquisition system controlled through the Sirenia Acquisition Software (Pinnacle Technology Inc.). The raw data was scored using Sirenia Sleep (Pinnacle Technology Inc.). The recordings were binned in 4-s epochs that were assigned initial scores using automatic threshold scoring, followed by manual correction. Vigilance states were scored as previously described in other studies[26,32], and MAs were scored as wake epochs when brief 4–12 s-long cortical activation with a decrease in slow and delta power bands and no spindles occurred with a concomitant increase in EMG activity during NREMS. The power of several frequency bands within each epoch was computed from Sirenia and exported for further analysis in custom Python scripts: Slow (0.5–1.0 Hz), Delta (1.0–4.5 Hz), Sigma (10.0–15.0 Hz). Any existing annotations for TTL or stimulation annotations were also exported separately, with a synchronized time to the sleep scores. Based on visual inspection of the EEG signals from the two hemispheres during scoring, either both EEG signals were kept and averaged, or only EEG signals without artifacts were kept. All subsequent analyses for in vivo recordings were performed using custom Python scripts unless otherwise stated.

## FP signal preprocessing

The FP signal was acquired with a 12 kHz sampling rate and downsampled to 0.25 Hz, using scipy interpolate.interp1d to match the sampling rate of the EEG recordings. An envelope of 1 s was created from the downsampled FP signal, using scipy.interpolate.PchipInterpolator. Then, the enveloped signal was corrected for bleaching by computing the coefficients for a least-squares 2nd-degree polynomial that best fits the signal containing only the periods during NREM sleep using numpy.polyfit. Then the coefficients were used to compute the fitted values for the entire signal (polyfit of the signal). $\Delta F/F$ was then calculated as: (signal−polyfit of signal)/polyfit of signal. Standardization of the FP signal was carried out by z-scoring either individual sections of the signal or all NREM bouts, depending on the situation, where $x$ is an individual time point as: ($x$−mean of $\Delta F/F$ at NREMS)/SD of $\Delta F/F$ at NREMS.

## Cross-correlation of the FP and EEG signals

To compute the cross-correlation between the sigma ISO and CRH release during NREMS, the FP and EEG signals were pre-processed as described above. Then, both signals were filtered with an FIR lowpass filter of 0.025 Hz and 100 taps using the scipy signal and filtfilt packages. Consolidated NREMS bouts were detected by searching for continuous NREMS bouts longer than 300 s, from which the first 100 s were deleted to discard the transition from wakefulness to NREMS. Each pair of detected signals was centered by subtracting the mean activity of the signals and standardized using numpy.linalg.norm. For each pair of detected signals, the cross-correlation between them was computed using scipy.signal.correlate for the entire length of the pair, then aligned and averaged per mouse.

## Power spectral density analysis

To compute the power spectral density (PSD) for FP and EEG optogenetic manipulations, FP and EEG signals were pre-processed as described above. Consolidated NREMS bouts were detected by searching for continuous NREMS bouts longer than 300 s, from which the first 100 s were deleted to discard the transition from wakefulness. Then, all NREMS bouts were cut to a length of 200 s. The detected signals were centered by subtracting the mean activity of the signals and standardized using numpy.linalg.norm. For FP signals, any remaining polynomial trend was removed using obspy.signal.detrend.polynomial with a maximal 3rd degree that was applied using a Hann window with scipy.signal.windows.hann over the entire length of the signal. PSD of individual NREMS bouts was calculated using Welch's method with scipy.signal.welch with one segment that is the length of the NREMS bout. The average PSD per mouse was computed and normalized using the AUC of the full PSD with numpy.trapz (trapezoidal rule). For the PSD of the EEG sigma band, the mean values of 0.08–0.12 Hz were used to bring the 0.02 Hz peak to a baseline of 0, before normalization with the full PSD AUC.

## Comparison of GRAB sensors

To compare of peak-to-peak amplitudes of $GRAB_{CRH}$ and $GRAB_{CRH}$mut during NREMS, the signals were pre-processed as mentioned above, filtered with an FIR lowpass filter of 0.025 Hz and 100 taps using the scipy signal and filtfilt packages. Consolidated NREMS bouts of at least 150 s were detected, and the first 16 s and last 4 s were deleted to remove transitions. For each pair of signals, peaks, and valleys were detected using scipy.signal.find_peaks with at least 40 s between them. The absolute difference between adjacent peaks and valleys was computed and then averaged for each mouse to obtain a final peak-to-peak value.

## Optogenetic manipulations and EEG/EMG recordings

For optogenetic excitation of CRH release, CRH-IRES-Cre × Ai27D male mice were implanted with EEG/EMG electrodes as described above. Two cannulas (BASi Research Products) were implanted bilaterally over the TRN, following the same coordinates as stated for the implantation of optic fibers. For optogenetic inhibition of CRH release, CRH-IRES-Cre male mice were injected with retroAAV2-Ef1a-DIO-PPO-Venus (PPO, $9.4 \times 10^{12}$ GC/ml; 300 nl) or the control virus AAVrg-Ef1a-DIO-EYFP (EYFP, $2.4 \times 10^{13}$ GC/ml; 300 nl) bilaterally into TRN (AP: −1.1; ML: ±2.36; DV: −3.47) as described above. The EEG/EMG electrodes and optic cannula implants followed the same procedure as described above, performed in the same session as the viral injection. For optogenetic manipulation of CRHR1 KD mice, male mice from CRH-IRES-Cre × Ai27D of 7 weeks were injected stereotactically as described in general surgery procedures with a control virus AAV9-hSyn-GFP (GFP, $2.7 \times 10^{11}$ GC/ml; 300 nl) or with AAV9-S5E2p-EGFPm-CRHR1-shRNAmir (shRNA, $1.26 \times 10^{13}$ GC/ml; 300 nl) bilaterally into TRN (AP: −1.1; ML: ±2.36; DV: −3.47). The full sequence of the shRNAmir is provided in Supplementary Data 1.

After 1 week of recovery from surgery for ChR2 and PPO mice or 3 weeks after recovery for the CRHR1 KD ChR2 manipulation, optic fibers connected to LEDs (200 μm diameter, 465 nm peak wavelength, Pinnacle Technology Inc.) were inserted into the cannulas and plugged into the preamplifier. After a week of habituation to the EEG/EMG recording cages, 2 days of undisturbed sleep were recorded as a baseline, followed by closed-loop optogenetic photostimulation on the following day for 4 h from 7 am to 11 am. Closed-loop optogenetic stimulation was designed and performed using the FeedbackPro module of the Sirenia Acquisition software. For the CRHR1 KD ChR2 manipulation, 5 weeks passed in total before the optogenetic stimulation, to allow viral expression and internalization of the remaining CRHR1 from the plasma membrane.

For the ChR2 experiments, including the CRHR1 KD ChR2 groups (GFP and shRNAmir), the stimulation protocol delivered blue light at 10 Hz for 50 s with a pulse width of 5 ms and a 55–60 mW/mm$^2$ power density measured at the fiber tip. For PPO experiments, the stimulation delivered blue light at 10 Hz for 50 s with a pulse width of 10 ms and a 45–55 mW/mm$^2$ power density measured at the fiber tip. For both manipulations, closed-loop rules were based on thresholds for the power density of sigma (10–15 Hz) and slow (0.5–1 Hz) oscillations, to detect and initiate the stimulation selectively during NREM sleep. The thresholds were adjusted for each mouse to trigger stimulations on average on sigma peaks, with a minimum delay of 50 s between stimulations.

At the end of the experiments, the brains were collected as described in the surgical procedures to assess opsin and/or viral expression and placement of optical fibers. The EEG recordings were scored and pre-processed as described in sleep scoring above. Two baseline days were always compared with 1 day of optogenetic stimulation. The processing steps described below were applied separately to the baseline days, and then the values were averaged to obtain one baseline value per mouse to compare with the optogenetic stimulation.

## Sleep bout analysis

The scoring and pre-processing of the EEG recordings were performed as described in the sleep scoring above. To distinguish MAs from wake epochs, the exported numerical sleep scores were modified with an algorithm we designed to mark MAs epochs with a different number only when continuous wake epochs do not exceed 12 s and are preceded by at least 16 s of NREM sleep and followed by at least 4 s of NREMS, in any other situation the wake epoch scores were not modified. Continuous sequences of the same sleep score, regardless of MAs for NREMS of at least 4 s, were detected for each recording. The number of MA events was defined similarly as continuous sequences of epochs marked as MA during NREM sleep. Several parameters from the detected bouts and events were computed as: average bout length (mean of the length of all bouts from the same vigilance state); % state/hour (the sum of the length of all bouts from the same vigilance state normalized by the length of the recording expressed as a percentage); bout count/hour (total number of detected bouts for the same vigilance state expressed as a normalized value per hour); MAs/hour of NREMS (total number of MA events normalized for the total time spent in NREMS expressed per hour of NREMS).

## Quantification of relative powers

Relative values for each power band were normalized as their percentage contribution to the full EEG spectrum across the entire recording as epoch power density/mean of full EEG band power density for the entire recording. Continuous sequences of the same sleep score, regardless of MAs for NREM sleep of at least 16 s, were detected for each recording. Relative powers were averaged per bout and then by categories of vigilance states per mouse.

For the analysis of the sigma surge at the transition to REMS, sigma power density was filtered with a lowpass of 0.025 Hz to reveal the ISO of sigma, as described in the cross-correlation of FP and EEG signal, and normalized as % relative power of the full EEG density. Based on the sleep score, the transitions to REMS were detected as continuous sequences of NREMS for at least 5 epochs, followed by continuous sequences of REMS for at least 5 epochs, regardless of MAs. The averages of the sigma oscillation at the transition to REMS were computed for each mouse for the baseline days and the stimulation day. The average sigma oscillation at the transition to REMS for each mouse was upsampled to 10 Hz to improve the time precision of the latency, and the highest amplitude peak within the last 50 s of NREMS prior to the first REMS epoch was detected using

scipy.signal.find_peaks. The amplitude of the peak and the latency to the first REMS epoch were computed for each mouse.

### Ex vivo electrophysiology

Mice were anesthetized with 5% isoflurane in carbogen (95% $O_2$/5% $CO_2$) and euthanized by decapitation. Whole-cell patch-clamp experiments were performed in 250 μm-thick acute horizontal slices containing the TRN, sliced with a vibratome (Campden Instruments, Loughborough, UK) in oxygenated (95% $O_2$/5% $CO_2$) ice-cold modified artificial cerebrospinal fluid (aCSF) containing (in mM): 105 sucrose, 65 NaCl, 25 $NaHCO_3$, 2.5 KCl, 1.25 $NaH_2PO_4$, 7 $MgCl_2$, 0.5 $CaCl_2$, 25 glucose, and 1.7 L(+)-ascorbic acid. Slices recovered for 1 h at 35 °C in standard aCSF containing (in mM): 130 NaCl, 25.9 $NaHCO_3$, 2.48 KCl, 1.25 $NaH_2PO_4$, 1.2 $MgCl_2$, 2 $CaCl_2$, 18 glucose, and 1.7 L(+)-ascorbic acid, complemented with 2 sodium pyruvate and 3 myo-inositol. In the recording chamber, the slices were superfused with standard oxygenated aCSF at RT. Visually identified TRN neurons were patched using borosilicate glass pipettes (2–4 MΩ), pulled at DMZ puller (Zeitz-Instruments) and filled with a solution containing (in mM): 130 KGluconate, 10 KCl, 10 HEPES, 0.2 EGTA, 10 phosphocreatine, 4 Mg-ATP, 0.2 Na-GTP (290–300 mOsm, pH 7.25). The liquid junction potential of −10 mV was not taken into account.

Burst and tonic firing was elicited in the current-clamp configuration in response to 2 s current injection steps ranging from 25 to 225 pA in increments of 25 pA while cells were held at −70 mV with direct current injection. An online bridge-balance correction was applied. Different concentrations of CRH (10–500 nM) were diluted in aCSF and bath-applied for 10–15 min.

For replication of this effect using endogenous CRH release in CRH-IRES-Cre × Ai27D mice, ChR2 was activated with a blue LED (465 nm) using a train stimulation of 20 Hz, 5 ms pulse width, for 22 s. To test the involvement of CRHR1, CRH was applied in the presence of the selective CRHR1 antagonist NBI35965 (3 μM, Tocris).

To monitor excitatory synaptic responses during optogenetic train stimulation of CRH release in CRH-IRES-Cre × Ai27D mice, the $GABA_AR$ antagonist picrotoxin (0.1 mM) was added to the aCSF, and cells were held at −70 mV by direct current injection in the current-clamp mode. For inhibitory synaptic currents, TRN neurons were voltage-clamped at −60 mV with an intracellular solution containing (in mM): 120 CsCl, 8 NaCl, 10 HEPES, 0.2 $MgCl_2$, 0.2 EGTA, 10 phosphocreatine, 2 Mg-ATP, 0.2 Na-GTP (290–300 mOsm, pH 7.2). AMPARs and NMDARs were blocked with DNQX (10 μM) and D, L-APV (50 μM), respectively.

Voltage-clamp recordings of $Ca_V3$ and SK2 currents were performed with automated leak subtraction in the presence of the $Na_V$ channel blocker tetrodotoxin (500 nM). For recordings of biphasic $Ca_V3$-SK2 currents, TRN neurons were voltage-clamped at −75 mV with an intracellular solution containing (in mM): 140 $KMeSO_4$, 10 KCl, 10 HEPES, 0.1 EGTA, 10 phosphocreatine, 2 Mg-ATP, 0.2 Na-GTP (290–300 mOsm, pH 7.2). A 1-s long depolarizing step of 40 mV was provided every 20 s. Charge transfer was measured instead of peak current amplitude because biphasic nature of the currents. In these cases, the $Ca_V3$-mediated component is truncated by the SK2-mediated component, preventing the accurate identification of peak currents. Negative and positive charge transfer components were quantified after zeroing mean current traces per minute relative to the steady-state current value at the end of the depolarizing step.

SK2-mediated tail currents were recorded from TRN neurons patched with the same intracellular solution and depolarized from −50 mV to +20 mV for 100 ms. The tail current area and decay time were measured in mean current traces per minute within a 500-ms temporal window, starting 15 ms after the end of the depolarizing step. For SK2-mediated tail currents, peak identification is inherently undefined, as tail currents decay gradually rather than displaying a distinct peak. Additionally, at the offset of the large depolarizing step,

abrupt voltage transitions can introduce signal distortions, further complicating maximal current measurements.

To isolate $Ca_V3$-mediated $Ca^{2+}$ currents, TRN neurons were voltage-clamped at −75 mV with an intracellular solution containing (in mM): 120 CsGluconate, 10 CsCl, 10 HEPES, 5 EGTA, 10 phosphocreatine, 4 Mg-ATP, and 0.2 Na-GTP (280–290 mOsm, pH 7.3). A single 1-s depolarizing step to −35 mV or three repetitive 100-ms steps at 5 Hz were applied every 20 s. Peak currents were quantified as the maximal amplitude relative to the steady-state current at the end of the depolarizing step.

Electrophysiological signals were acquired through a Digidata1550A digitizer, amplified through a Multiclamp 700B amplifier, sampled at 20 kHz, and filtered at 10 kHz using Clampex 10.6 (Molecular Devices). Analysis was performed in Clampfit 10.6 (Molecular Devices).

### Statistical analysis

Statistical comparisons were performed in GraphPad Prism v. 10.3.1 and Python v. 3.11.3. All data are represented as the mean ± standard error of the mean (SEM) or kernel density estimation (KDE) for violin plots with median and upper/lower quartiles. The normality of the data distribution was assessed using the Shapiro–Wilk test. For most datasets, the experimental designs allowed for paired statistical analyses using two-tailed paired $t$-tests for two groups. Repeated measures two-way or one-way analysis of variance (ANOVA) was used for multiple measurements under different conditions, followed by post-hoc Holm–Šídák's multiple comparisons test. For unpaired comparisons, two-tailed unpaired $t$-tests were used, or for comparing more than three groups, one-way ANOVA was used. When data were not normally distributed, Mann–Whitney or Kruskal–Wallis tests were used. Statistical significance was taken with a critical probability of $\alpha < 0.05$ with corrected α thresholds for multiple comparisons using Bonferroni's correction method. The correlation $r$ values were computed using Pearson's correlation coefficients. Each datapoint in the graphs represents an individual animal or cell, as indicated in the figure legends. The statistical details of each experiment can be found in Supplementary Table 1.

### Reporting summary

Further information on research design is available in the Nature Portfolio Reporting Summary linked to this article.

## Data availability

All processed data supporting the findings of this study are available within the Article, the Supplementary Information, and the accompanying Source data file. Due to the large size of the raw datasets, these are available from the corresponding authors upon request. No restrictions apply to the data presented in this study. Source data are provided with this paper.

## Code availability

Custom codes developed in the study, along with example datasets, are available here: https://doi.org/10.6084/m9.figshare.29195285.

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

## Acknowledgements

We thank Dr. Anita Lüthi for helpful discussions and expert feedback on the study and manuscript. We also thank Dr. Michael Bruchas and Dr. Tallie Baram for their support in providing access to key tools essential for this research. We are grateful to Jocelyn Grosse and Isabelle Guillot de Suduiraut for technical assistance. We also acknowledge the EPFL research facilities that facilitated the experimental procedures, the Center of PhenoGenomics for the rodent husbandry, and the BioImaging and Optics Core Facility for training and advice on microscopy imaging. This study was supported by grants from Syn2Psy, funded through Marie Skłodowska-Curie actions (Grant agreement ID: 813986) to C.S. and S.A.,

Novartis Foundation for medical-biological research (Funds 534043) to S.A., intramural funds from EPFL to C.S., and NIH U01 NS128537 to B.C.

## Author contributions

S.A., L.C., and C.S. conceived the project and designed the experiments. S.A. and C.S. obtained the financial support and supervised the project. L.C., S.A., and O.Z. carried out experiments and analyzed data, curated data visualization, and wrote the manuscript. D.K. and B.C. developed and provided viral tools for optogenetic inhibition. All authors edited the manuscript.

## Competing interests

The authors declare no competing interests.
