## [Transparent Peer Review file · Nature Communications]

Corticotropin-releasing hormone modulates NREM sleep consolidation through the thalamic reticular nucleus

Corresponding Author: Dr Simone Astori

Version 0:

Reviewer comments:

Reviewer #1

(Remarks to the Author)

The manuscript by Cumpana and colleagues investigates the role of CRF neurotransmission and CRF-R1 signaling in the thalamic reticular nucleus (TRN) in regulating non-REM sleep. Using state-of-the-art and multidisciplinary approaches, the investigators tested the hypothesis that CRF acts as a pro-arousal neuropeptide neurotransmitter to modulate NREM during “fragility periods” to promote microarousals. The study demonstrates that CRF-R1 is expressed on and regulates the burst probability of low-threshold firing PV (and SOM) neurons. It shows that CRF is released in the TRN during NREM fragility points. Further, optogenetic manipulations of CRF terminals can bi-directionally modulate microarousals (stimulation increases; inhibition decreases) and sigma power. Finally, they demonstrate that CRF modulates microarousals via CRF-R1s on PV neurons in the TRN using a combination of optogenetics and shRNA directed toward CRF-R1. Overall, this was an excellent study. It is both conceptually and technically innovative. I was very impressed overall with the level of rigor. I was especially impressed with the beautiful electrophysiological studies. I have one major concern that needs to be addressed prior to publication, likely with some additional experiments. I also have several moderate/minor concerns that should be thoroughly addressed as well in order for the manuscript to be suitable for publication.

I did think it was a real missed opportunity to not look at both males and females for the in vivo studies given clear sex biases in several disorders where sleep disturbances are common features. I would strongly encourage the investigators to pursue this in the future.

Major concern:

1. My major concern is with the use of the GRAB-CRH sensor. While use of this tool certainly adds to the innovation of the study, since it is such a novel tool, it also requires some additional validation. The GRAB-CRH 3.0 from Yulong Li's lab has been validated and recently published. This was done in several ways including through use of CRISPR-Cas9 mutagenesis for Crh. However, I could not find any reference for this mutant form of the GRAB-CRH sensor from WZ Biosciences and I could not find the details for how this was generated (these should be provided). I think further validation is necessary. One way to do this would be in an ex vivo slice preparation of the TRN where the investigators would show differences in bath application of CRF with and without CRF-R1 antagonist for the WT GRAB-CRH sensor and the mutant GRAB-CRH. This should be pretty straight forward for the investigators to do with their available skill sets.

As an aside, it would be cool to show that the optogenetic stimulation used for the NREM studies caused the release of CRH using the GRAB-CRH 3.0 using an ex vivo slice prep. However, for these experiments, you could use an Ai27D cross. You would need to use a red-shifted opsin. This is not a necessary experiment; it would just be interesting to pursue this in the future.

Moderate/Minor concerns:

1. In the results, make note that the in vivo experiments were only done in males.

2. Figure 1:

a. I would suggest combining all or part of Supp Figure 1 into Figure 1. There is room, and I found that the low power image showing the PV neurons was very useful.

b. More details need to be provided regarding the density measurements. I don't quite understand why it is in arbitrary units instead of number of puncta. Are the investigators using the particle counter feature in Fiji? I would like to understand a bit

more about the analysis.

3. Figure 2:

- a. Found this figure a little confusing. I think it would be remedied by labeling with the viral vector being displayed rather than just CRH+ and also indicate that this is all in CRH-IRES-Cre mice on the figure. I would also indicate in the figure the coronal plane relative to bregma for each panel each on the panel or in the figure legend.
- b. If you have the room, I think it would be nice to include the viral vector information in the Results as well as methods, so it is easier for people to follow.
- c. A little concerned that for the ZID there is only an N= 1. Perhaps necessary to increase the N to ensure reproducibility.

4. Figure 4/7: Include high power images/perhaps with co-labeling with synaptophysin to better show synaptic terminal labeling with Chr2 around the cells in the TRN.

5. In the Discussion, the authors should include discussion about the regional differences in tropism, particularly with retrograde AAV (Tervo et al., 2016, Neuron), that should be considered when interpreting the data in Figure 2.

Reviewer #2

(Remarks to the Author)

In this work, Cumpagna et al. studied the dynamics of free Corticotropin-Releasing Hormone (CRH) in the thalamic reticular nucleus (TRN) during sleep, with a particular focus on its function during non-rapid-eye movements (NREM) sleep. Using state-of-the-art techniques, they showed a role of CRH receptors type 1 (CRHR1) in parvalbumin (PV) interneurons as key for regulating brain sleep rhythms and sleep fragmentation. They additionally explored the potential cellular mechanisms underlying these effects. Natural fluctuations of CRH are anticorrelated to sigma activity on an infraslow, close-to-minute timescale. The CHR release in the TRN reduces sigma activity and increases microarousals (MA) during NREM sleep, probably due to CHRR1-mediated decreases in low-threshold bursting of TRN neurons. In contrast, inhibition of CHR release in this brain area reduces MA density and increases activity in the cortex.

The methodological design and technical and analytical approaches are current and appropriate for pursuing the hypothesis tested, and the results carefully support the conclusions. Overall, the results support their interpretation of a role of CHR in the modulation of continuity in addition to the existing evidence for other neuromodulators. These findings have the potential to significantly impact our understanding of sleep regulation and its implications for stress-related and sleep disorders. Needless to say, it has potential significance for studying central-autonomic interactions during sleep. I have limited concerns about suggestions that might increase the significance, clarity, and scope of the study:

Stimulation protocol and parameters: What is the rationale behind the 50 seconds of optogenetic manipulations and the 10 Hz stimulation? Previous studies have assessed the dependency of the behavioral response to different stimulation frequencies during sleep for other related systems (e.g., Carter et al., 2010). Additionally, if understood correctly, the closed-loop optogenetic manipulations started mainly at the peak of sigma activity and lasted 50 seconds. Given the fluctuation of CRH within the same timescale (one of your main interesting results), the question of how the response depended on the stimulation protocol and how long it would maintain a sustained effect for at least two periods of such oscillations (>100 s) or even throughout the entire NREM bout. Longer manipulation protocols could allow you to put in evidence the effects of the CHR (and glutamate, see below) already in the raw data of Fig. 4b (low) and 5c (low). Additionally, it would be ideal to depict the delta dynamics, given the interesting result of the optogenetic manipulations.

This might be of relevance given the time dependency similar effects in other systems (Reference 32 in manuscript: Osorio-Forero et al., 2021- Figure 4) or depleting molecular content for "fast" stimuli (Silverman et al., 2024). If possible, a single or couple of subjects demonstrating the frequency and protocol dependency of the results can resolve these questions within the framework of the study. And could further resolve future experimental considerations for the field.

Similarly, the results from the supplementary Figure 2 on the effects of the optogenetic manipulations can be best answered with longer stimulation protocols. Without such control experiments, the authors might arrive at contrasting results such as those in lines 153-154 (reduction of sigma without affecting its periodicity). These contrasts might be an artifact of the protocol.

Sigma power as a proxy for spindle density: Although sigma power is among the best representations of spindle density, the specific detection of sleep spindles and quantification of spindle density can strengthen the study's conclusions (as in line 159).

Corelease glutamate and CRH: One of the major results is the corelease of glutamate with the CRH terminals stimulation. Intriguingly, glutamate in PV cells does not affect sleep, as shown in the results in Figure 7 i-k. What are the author's interpretations of this result? Would manipulation of glutamate signaling be sufficient or necessary to modulate any of these responses? Although evidently out of the scope of the study, it would be at least recommended that the discussion on the matter be strengthened. Alternatively, if time allows, pharmacological/optogenetic manipulations of such pathways could significantly increase the significance of the study.

Similarly, what effects does it have on the CRHR1 KO in the TRN? Is sleep rhythms, architecture, or fragmentation affected in any way by this manipulation? Comparing the baseline condition of Animals in Figure 7h-k to the baseline of their control group would be sufficient to support/contrast the claims from the results in Figure 5 on CRH inhibition during NREM sleep.

Cellular mechanisms of CRHR1 on calcium burst: From a nonexpert perspective on in-vitro electrophysiology, the outstanding additional explanatory level of the cellular mechanisms represents a major contribution to the field. I am also curious about the underlying mechanisms of suppressing calcium bursting in TRN neurons. Is it via partial depolarization of the TRN cells or by internal modulation of the CaV3 Ca2+? A more detailed explanation or voltage clamp examples of such effects might promote and facilitate further discussions in the field.

Sleep stages and power spectra: Representative epochs of the sleep stages and average (or grand average) power spectra for each sleep stage (with the different manipulations) and not only the average sigma and/or delta power are often ideal to represent the effects in overall brain activity. Supplementary information containing such figures gives the reader additional support for the quality and strength of the results. These can be done in log-linear or log-log spectra, which might also be enough to support your claims to skeptical readers.

REM sleep transitions: Although the latency to REM sleep is an appropriate proxy for the CRH effect on NREM-to-REM transitions, the most convincing metric to support some of the conclusions is the actual NREM-to-REM transition density per [unit of time (min or hr)] of NREM sleep. Additionally, the optogenetics protocols raise the question of how much the NREM-to-REM transition latency effects depend on a post-manipulation rebound. An average transition density before, during, and after the optogenetic manipulation(s) might help clarify.

Similarly, what is the rationale behind the pre-REM sigma peak analyses of Figures 4f and 5g? Given the result of reduced averaged sigma activity, these accessory results seem redundant or do not seem to provide additional information. Instead, they raise more questions about the stimulation protocol's effect and post-manipulation(s) rebound. It's important to clarify the relevance of these results and how they contribute to the study's conclusions. Finally, how much does the minimum bout length of REM sleep (5 epochs, 20 s) affect the NREM-to-REM transition result? Would it be the same for shorter (2 to 3 epochs)?

Minor comments

- In lines 56-57: If needed, the statement "Notably, 56 NREMS fragility periods have a higher probability for the occurrence of MAs" might be supported by Cardis et al., 2021.
- In line 446, AP, ML, and DV for the coordinates are missing.
- Figure titles sometimes refer to the result and other times to the methods. As in the case of the results, one suggestion is to homogenize one approach or another for the figures.
- In Figure 3e, grey lines represent the peaks and troughs of the sigma activity, yet they are not described in the legend.
- Figure 3 misses the legend for 3d.
- For the cross-correlation analysis, it is often helpful to also depict the lag of the peak correlation and not only the correlation at zero lag.
- For the list of features in lines 581-587, I suggest using ";" instead of "-" to separate the features, as sometimes "-" could be mistaken for a minus, disrupting the reading fluidity.

Carter, M. E., Yizhar, O., Chikahisa, S., Nguyen, H., Adamantidis, A., Nishino, S., ... & De Lecea, L. (2010). Tuning arousal with optogenetic modulation of locus coeruleus neurons. *Nature Neuroscience*, 13(12), 1526-1533.

Silverman, D., Chen, C., Chang, S., Bui, L., Zhang, Y., Raghavan, R., ... & Dan, Y. (2024). Activation of locus coeruleus noradrenergic neurons rapidly drives homeostatic sleep pressure. *bioRxiv*, 2024-02.

Cardis, R., Lecci, S., Fernandez, L. M., Osorio-Forero, A., Chu Sin Chung, P., Fulda, S., ... & Lüthi, A. (2021). Cortico-autonomic local arousals and heightened somatosensory arousability during NREMS of mice in neuropathic pain. *Elife*, 10, e65835.

Version 1:

Reviewer comments:

Reviewer #1

(Remarks to the Author)

Thank you to authors for their patience and thorough response in addressing all my concerns. The authors have now added several additional experiments and added methodological details and discussion that improve study and quality of the manuscript. It is now suitable for publication. Congratulations on an excellent study!

Reviewer #2

(Remarks to the Author)

The manuscript by Cumpana et al. is an excellent example of a hypothesis-driven approach in systems neuroscience, particularly in the field of sleep. They studied the network, cellular, and functional mechanisms of corticotropin-releasing hormone (CRH) in the Reticular Thalamic Nucleus (TRN) and its role in sleep architecture, microarchitecture, and spontaneous arousability in NREM sleep. They described close-to-minute fluctuations in CRH in the TRN, similar to those of other monoaminergic systems, and related to electrophysiological markers of arousability on this timescale. They also demonstrated the sufficiency of this molecule in modulating the fragility of NREM sleep through optogenetic and genetic

approaches. Furthermore, through rigorous *ex vivo* experiments, they demonstrated that CRH acts on the spindle-generating PV cells of the TRN by progressively attenuating Cav3 currents. I praise the excellent figures throughout the manuscript, which effectively depict the significant results in a clear and accessible manner, even with the example/representative traces. In this revised version of the manuscript, the authors addressed all my concerns in detail. I do not have further relevant concerns, but I have included my response to some of the raised points below.

- Through their explanation, I am less worried, although still curious, about the lack of effect on the infraslow fluctuations in sigma that accompany the reduction in sigma after optogenetic stimulation. However, in light of the outstanding, long-lasting impact of CRH in TRN cells (as shown in the *ex vivo* experiments), this raises questions on the nature of the sigma activity in comparison with individual spindles or other electrophysiological characteristics of the signal (e.g., aperiodic activity).

- In this line of thought. I agree that local field potentials (particularly those of the somatosensory cortex) are best suited for spindle analysis. However, the EEG has also been used in the past to study spindles in rodents (e.g., Mölle et al., 2009; Lacroix et al., 2018; Fernandez et al., 2018). Although outside the scope of this study, it would be interesting to address these differences in the future. I share a simple script in this response that may be useful for quantifying spindles in NREM sleep for future reference.

- Concerning the stimulation protocols. I apologize for the lack of precision in my comment regarding the stimulation frequencies, as I was curious about the lower frequencies (not higher). However, first, it is still very valuable to highlight the potential ceiling effect of the 10 Hz stimulation with the new control experiment at 20 Hz. I agree with the authors in emphasizing the complexity and scope of such experiments, which would be interesting to conduct in other studies, particularly those examining specific firing patterns of CRH+ cells. Co-release mechanisms are a fascinating and complex topic worth tackling across neural systems.

- I had a previous concern about what turned out to be one of the most exciting results. Namely, the NREM-to-REM sleep transitions in the *in vivo* manipulations. The creative analysis of REM latency in comparison to the sigma peak, or in addition to the transition density (TTR in the manuscript), highlights the importance of the intermediate state and the complexity of the neurochemical state, particularly in the context mentioned by the authors, namely stress-related disorders.

References

- Mölle M, Eschenko O, Gais S, Sara SJ, Born J. The influence of learning on sleep slow oscillations and associated spindles and ripples in humans and rats. *Eur J Neurosci* 29: 1071–1081, 2009. doi:10.1111/j.1460-9568.2009.06654.x.
- Lacroix MM, de Lavilléon G, Lefort J, El Kanbi K, Bagur S, Laventure S, Dauvilliers Y, Peyron C, Benchenane K. Improves sleep scoring in mice reveals human-like stages. *bioRxiv* 489005, 2018. doi:10.1101/489005.
- Fernandez LM, Vantomme G, Osorio-Forero A, Cardis R, Béard E, Lüthi A. Thalamic reticular control of local sleep in mouse sensory cortex. *eLife* 7: e39111, 2018. doi:10.7554/eLife.39111.

Manuscript Number: NCOMMS-24-63173

Corticotropin-releasing hormone modulates NREM sleep consolidation through the thalamic reticular nucleus

Cumpana et al.

Replies to Reviewers' comments:

Reviewer 1

Reviewer #1: The manuscript by Cumpana and colleagues investigates the role of CRF neurotransmission and CRF-R1 signaling in the thalamic reticular nucleus (TRN) in regulating non-REM sleep. Using state-of-the-art and multidisciplinary approaches, the investigators tested the hypothesis that CRF acts as a pro-arousal neuropeptide neurotransmitter to modulate NREM during “fragility periods” to promote microarousals. The study demonstrates that CRF-R1 is expressed on and regulates the burst probability of low-threshold firing PV (and SOM) neurons. It shows that CRF is released in the TRN during NREM fragility points. Further, optogenetic manipulations of CRF terminals can bi-directionally modulate microarousals (stimulation increases; inhibition decreases) and sigma power. Finally, they demonstrate that CRF modulates microarousals via CRF-R1s on PV neurons in the TRN using a combination of optogenetics and shRNA directed toward CRF-R1. Overall, this was an excellent study. It is both conceptually and technically innovative. I was very impressed overall with the level of rigor. I was especially impressed with the beautiful electrophysiological studies. I have one major concern that needs to be addressed prior to publication, likely with some additional experiments. I also have several moderate/minor concerns that should be thoroughly addressed as well in order for the manuscript to be suitable for publication.

Authors' reply: We thank the Reviewer for the appreciation of our study and for the specific comments below that helped us improving our manuscript.

Specific comments:

Reviewer #1: I did think it was a real missed opportunity to not look at both males and females for the *in vivo* studies given clear sex biases in several disorders where sleep disturbances are common features. I would strongly encourage the investigators to pursue this in the future.

Authors' reply: We also thank the Reviewer for acknowledging this important topic that we agree is a priority for future studies, which we discuss in the limitations of our study. To expand to our current abilities, we have added a sex-based comparison for our *ex-vivo* analysis of TRN bursting in response to CRH bath application (Supplementary Fig. 7), showing no difference in the change in the number of bursts with CRH application between male and female mice.

Supplementary Fig. 7. CRH affect TRN bursting in both sexes without altering the membrane potential

a Example voltage traces from TRN neurons in a male and a female mouse before (Bsl) and after (CRH) application of 500 nM CRH and quantification of the %change in the number of bursts, showing no difference between sexes.

Reviewer #1:

Major concern:

1. My major concern is with the use of the GRAB-CRH sensor. While use of this tool certainly adds to the innovation of the study, since it is such a novel tool, it also requires some additional validation. The GRAB-CRH 3.0 from Yulong Li's lab has been validated and recently published. This was done in several ways including through use of CRISPR-Cas9 mutagenesis for *Crh*. However, I could not find any reference for this mutant form of the GRAB-CRH sensor from WZ Biosciences and I could not find the details for how this was generated (these should be provided). I think further validation is necessary. One way to do this would be in an *ex vivo* slice preparation of the TRN where the investigators would show differences in bath application of CRF with and without CRF-R1 antagonist for the WT GRAB-CRH sensor and the mutant GRAB-CRH. This should be pretty straight forward for the investigators to do with their available skill sets.

As an aside, it would be cool to show that the optogenetic stimulation used for the NREM studies caused the release of CRH using the GRAB-CRH 3.0 using an *ex vivo* slice prep. However, for these experiments, you could use an Ai27D cross. You would need to use a red-shifted opsin. This is not a necessary experiment; it would just be interesting to pursue this in the future.

Authors' reply: We agree with the Reviewer that caution is necessary in utilizing novel techniques. We have enquired with the Yulong Li lab and have received confirmation that the virally encapsulated GRAB_{CRH}3.0 used in our study (AAV9-hSyn-CRF3.0 from WZ Biosciences) is the same GRAB_{CRH} construct that is described and validated in their published study¹, which has been renamed since the study from GRAB_{CRH}1.0 to GRAB_{CRH}3.0 with no modification to the structure. The same applies to the mutant version of the GRAB_{CRH}, which is also the same version of the construct as described in their paper.

Additionally, we believe it is important to test the *ex-vivo* validation method suggested by the Reviewer to test its replicability under our conditions. We have performed new experiments to provide such validation. The data are presented in the Supplementary Fig. 2 of the revised manuscript, which is included below.

Supplementary Fig. 2. Ex-vivo validation of the GRAB_{CRH} sensor

a Left, example images of basal fluorescence in TRN cells infected with GRAB_{CRH} and GRAB_{CRH}mut (scale bar = 100 μm). Right, average responses of TRN neurons to bath-applied CRH (100 nM for GRAB_{CRH} and 500 nM for GRAB_{CRH}mut), expressed as % change in fluorescence compared to baseline (ΔF/F₀). An additional experimental series was included in which GRAB_{CRH}-infected slices were preincubated with the CRHR blocker αH-CRH (100 nM).

b Quantification of max ΔF/F₀ in the three groups, validating GRAB_{CRH} as a reporter of CRH.

This experiment confirms through *ex-vivo* live fluorescent imaging the results described in Wang et al., 2023¹ (relevant graph from Fig. 5 added below), where they elicit endogenous CRH release with electrical stimulation of central amygdala axons and measure fluorescence with two-photon *ex-vivo* live imaging in acute slices. Our validation experiment confirms the high reactivity of GRAB_{CRH} to CRH, which can be significantly decreased by using the competitive antagonist αH-CRH. The sensitivity to CRH is completely abolished in the GRAB_{CRH}mut, as described in greater detail in their paper¹.

Lastly, as mentioned as well by the Reviewer, it would be useful in future experiments to measure concomitantly the CRH release through GRAB_{CRH} in response to the optogenetic stimulation, which we also address in our limitations. Such measurements would not only serve as additional validation for the experiments but also expand the possibility of understanding the importance of specific frequency-band oscillations of neurotransmitter release and their relationship to EEG oscillations during sleep.

Reviewer #1:

Moderate/Minor concerns:

1. In the results, make note that the *in vivo* experiments were only done in males.

Authors' reply: As recommended by the Reviewer, we have clarified at the beginning of the *in vivo* section that recordings were acquired in males. We further discuss this issue as the first point in the section 'Limitations of the study'.

2. Figure 1:

a. I would suggest combining all or part of Supp Figure 1 into Figure 1. There is room, and I found that the low power image showing the PV neurons was very useful.

b. More details need to be provided regarding the density measurements. I don't quite understand why it is in arbitrary units instead of number of puncta. Are the investigators using the particle counter feature in Fiji? I would like to understand a bit more about the analysis.

Authors' reply: We also thank the Reviewer for helping us to clarify further the arrangement and reporting on Figure 1.

We have attached below the revised Figure 1, which now include panels from the previous Suppl. Fig. 1. The density measurements are indeed the number of puncta/ μm^2 and not the fluorescence intensity of the puncta, which would be reported in arbitrary units. This has now been corrected in the Figure; we thank the Reviewer for noticing this mistake.

The units are now reported as number of puncta/ μm^2 in the figure (see below). We have also added clarifying details to the relevant Methods section on the specific analysis used to obtain these measurements (see with red text below showing the new additions).

Fig. 1. CRHR1 mRNA is highly expressed in parvalbumin-positive TRN neurons

a Example image overview of RNAscope *in situ* mRNA hybridization stained for DAPI (blue), with parvalbumin (PV – magenta) highlighting the TRN and white squares indicating the corresponding regions shown as close-ups in hippocampal CA3 (**b**), TRN (**c**), and BLA (**d**), where CRHR1 mRNA can be seen as white puncta.

e TRN has significantly higher levels of CRHR1 mRNA compared to CA3 and BLA ($n = 11$).

f No significant difference was found between males ($n = 6$) and females ($n = 5$) in CRHR1 mRNA expression in TRN.

g Example image of *in situ* mRNA hybridization for parvalbumin (PV - magenta), somatostatin (SOM - green) and CRHR1 (white) mRNA labeling in TRN, showing the expected anatomical segregation into a PV + core and SOM+ shell in the sensory sector.

h Close-up of TRN, exemplifying a higher density of CRHR1 mRNA puncta in PV + neurons compared to SOM+ neurons.

i CRHR1 mRNA has a significantly higher expression in PV+ TRN neurons. Each point is the average density of CRHR1 mRNA in PV+ or SOM+ only TRN nuclei detected with DAPI of a mouse ($n = 11$).

Excerpt from Methods – mRNA quantification with RNAscope

“The acquired images were processed in ImageJ Fiji, using a script in Macro language. DAPI, PV, and SOM channels were segmented to create masks used in the quantification of the CRHR1 puncta, **which were detected using the maxima with a prominence of 35 after background subtraction** adjusted for the mean fluorescence -2 SD, as taken from the histogram of the fluorescence intensity distribution from each mouse. The density of CRHR1 mRNA puncta (**number of puncta/ total area of segmentation mask**) was determined for all segmentation masks and for the intersection between different masks, normalized by area, resulting in **number of puncta/ μm^2** . The resulting CRHR1 densities from multiple images were averaged per mouse.”

Reviewer #1:

3. Figure 2:

a. Found this figure a little confusing. I think it would be remedied by labeling with the viral vector being displayed rather than just CRH+ and also indicate that this is all in CRH-IRES-Cre mice on the figure. I would also indicate in the figure the coronal plane relative to bregma for each panel each on the panel or in the figure legend.

b. If you have the room, I think it would be nice to include the viral vector information in the Results as well as methods, so it is easier for people to follow.

c. A little concerned that for the ZID there is only an N= 1. Perhaps necessary to increase the N to ensure reproducibility.

Authors' reply: We thank the Reviewer for the suggestions on how to improve Figure 2, which we have followed up. The revised figure is attached below.

- In this revised figure, we have added a diagram in the left column for each injection protocol showing both the viral vector and the mouse line used in each case, along with the coronal plane relative to bregma for each image.
- In the main text, for consistency to make it easier for the reader to follow, we have added also the full name of the viral vectors in the results section. The viral information is presented in full in the Methods section.
- We agree with the Reviewer that, in order to increase the robustness of this experiment, the results for the anterograde confirmation of ZID as a CRH projection to the TRN should be replicated to reach a number similar to the other presented projections. In order to address this point, we have now performed further studies in which we injected additional CRH-IRES-Cre mice with the anterograde Cre-dependent AAV-GFP (AAV1-pCAG-FLEX-EGFP) in ZI, as done previously. In two mice in which the infection was well-targeted and restricted to the ZI (we had to discard others where there was slight misplacement), the virally infected CRH positive axons stemming from ZID were traced back to the TRN (see Fig. below, scale bars are 200 μ m), confirming our previous results and increasing the n to 3 in our revised manuscript.

Reviewer #1: 4. Figure 4/7: Include high power images/perhaps with co-labeling with synaptophysin to better show synaptic terminal labeling with Chr2 around the cells in the TRN.

Authors' reply: We thank the Reviewer for this suggestion and agree to the importance of understanding the overlap between Chr2-expressing terminals and the CRH-expressing terminals in the crossed transgenic mouse line CRH-IRES-Cre x Ai27D we have widely used throughout the study. To address this issue, we have now performed an immunohistochemical experiment in CRH-IRES-Cre x Ai27D mice to test the colocalization of the tdTomato-labeled Chr2 expressed under the CRH-Cre promoter with the presynaptic marker Chromogranin A, which labels large dense-core vesicles such as the CRH-containing vesicles, as synaptophysin is not often or unreliably found in CRH-containing dense core vesicles. In the supplementary Fig. 1 of the revised manuscript (see below), we present example images showing Chromogranin A puncta in CHR-releasing fibers in the CHR-axons in the TRN, BLA and PVN, along with a negative control for the Chromogranin A antibody. Quantification of immunoreactivity indicates comparable levels in TRN and the two other brain regions, in which CRH release is well documented.

We describe these finding in the Results section, second subsection, as follows:

To further support that CRH is released in the TRN, we used CRH-IRES-Cre x Ai27D mice, in which the TdTomato fluorescent reporter for Chr2 labels all CRH-positive neurons. We performed immunostaining for Chromogranin A, a marker of large dense-core vesicles previously used to show CRH release in the PVN². Quantification of Chromogranin A puncta within TdTomato-labeled fibers revealed comparable expression levels across the TRN, BLA, and PVN (Supplementary Fig. 1).

Supplementary Fig. 1. Large dense core vesicles are present in CRH-releasing fibers in the TRN

a-c Example confocal images showing immunoreactivity for Chromogranin A (ChromA), TdTomato (TdTom) and parvalbumin (PV), marking large dense core vesicles, CRH-releasing fibers and PV-positive cells, respectively, in TRN, BLA and PVN from a CRH-IRES-Cre X Ai27D mouse. Low magnification images (scale bar = 20 μm) are shown with higher magnification images (scale bar = 10 μm) of the marked area. Intensity of individual pseudo-coloured channels were adjusted before merging for visualization purpose.

d Negative control immunostaining for ChromA.

e Quantification of ChromA immunoreactivity in TdTom-positive fibers in the three brain regions.

Reviewer #1: 5. In the Discussion, the authors should include discussion about the regional differences in tropism, particularly with retrograde AAV (Tervo et al., 2016, Neuron), that should be considered when interpreting the data in Figure 2.

Authors' reply: We thank the Reviewer for bringing up an important point we have missed in our former discussion. While the rgAAV2 does not exhibit regional tropism in the classical sense, having a mostly uniform expression pattern throughout the CNS, Tervo et al., 2016³ do report potential issues at the level of specific circuits or neuronal populations in terms of sensitivity to rgAAV2 expression and retrograde transport. We have added a new paragraph, pasted below, in the limitations of our study, discussing this issue:

Additional studies are necessary to fully elucidate the identity of CRH-projecting regions to the TRN, particularly in light of the technical limitations associated with the genetic tools used in our study. Our approach, which involved Cre-dependent retrograde viral tracing or opsin expression in combination with the CRH-IRES-Cre mouse line, could potentially result in false-positive labeling due to persistent Cre expression in adulthood following transient CRH expression during early development⁴. While there is evidence supporting sustained CRH expression throughout development in regions such as the BLA⁵, MGM, and ZI^{4,6}, it remains to be verified whether CRH expression in the SGN and VLGPC is transient or persists into adulthood.

On the other hand, our tracing approach may not have captured all potential CRH sources projecting to the TRN. Notably, our experiments primarily targeted the sensory TRN, leaving CRH projections to

the anterior TRN unexamined. Moreover, the rgAAV2 viral construct used in our study has been reported to inefficiently transduce specific neuronal circuits and subpopulations, such as corticothalamic neurons and dopaminergic neurons in the substantia nigra³. To address these limitations, alternative viral tools with improved tropism will be necessary to identify fully the CRH-projecting inputs to the TRN.

Reviewer 2

Reviewer #2: In this work, Cumpana et al. studied the dynamics of free Corticotropin-Releasing Hormone (CRH) in the thalamic reticular nucleus (TRN) during sleep, with a particular focus on its function during non-rapid-eye movements (NREM) sleep. Using state-of-the-art techniques, they showed a role of CRH receptors type 1 (CRHR1) in parvalbumin (PV) interneurons as key for regulating brain sleep rhythms and sleep fragmentation. They additionally explored the potential cellular mechanisms underlying these effects. Natural fluctuations of CRH are anticorrelated to sigma activity on an infraslow, close-to-minute timescale. The CHR release in the TRN reduces sigma activity and increases microarousals (MA) during NREM sleep, probably due to CHRR1-mediated decreases in low-threshold bursting of TRN neurons. In contrast, inhibition of CHR release in this brain area reduces MA density and increases activity in the cortex.

The methodological design and technical and analytical approaches are current and appropriate for pursuing the hypothesis tested, and the results carefully support the conclusions. Overall, the results support their interpretation of a role of CHR in the modulation of continuity in addition to the existing evidence for other neuromodulators. These findings have the potential to significantly impact our understanding of sleep regulation and its implications for stress-related and sleep disorders. Needless to say, it has potential significance for studying central-autonomic interactions during sleep. I have limited concerns about suggestions that might increase the significance, clarity, and scope of the study:

Authors' reply: We thank the Reviewer for the appreciation of our study and for the specific comments below that helped us improve our manuscript.

Reviewer #2: Stimulation protocol and parameters: What is the rationale behind the 50 seconds of optogenetic manipulations and the 10 Hz stimulation? Previous studies have assessed the dependency of the behavioral response to different stimulation frequencies during sleep for other related systems (e.g., Carter et al., 2010). Additionally, if understood correctly, the closed-loop optogenetic manipulations started mainly at the peak of sigma activity and lasted 50 seconds. Given the fluctuation of CRH within the same timescale (one of your main interesting results), the question of how the response depended on the stimulation protocol and how long it would maintain a sustained effect for at least two periods of such oscillations (>100 s) or even throughout the entire NREM bout. Longer manipulation protocols could allow you to put in evidence the effects of the CHR (and glutamate, see below) already in the raw data of Fig. 4b (low) and 5c (low). Additionally, it

would be ideal to depict the delta dynamics, given the interesting result of the optogenetic manipulations.

This might be of relevance given the time dependency similar effects in other systems (Reference 32 in manuscript: Osorio-Forero et al., 2021- Figure 4) or depleting molecular content for “fast” stimuli (Silverman et al., 2024). If possible, a single or couple of subjects demonstrating the frequency and protocol dependency of the results can resolve these questions within the framework of the study. And could further resolve future experimental considerations for the field.

Authors’ reply: We thank the Reviewer for highlighting several relevant points related to our protocol design. We fully agree that it is important to detail them clearly.

In reply to the specific Reviewer’s points, our main rationale for choosing the 50 s optogenetic stimulation paradigm is based on a previous study on noradrenaline that showed a time/phase dependency of the spindle modulating effects of noradrenaline release in TRN⁷, as mentioned by the Reviewer as well. However, compared to noradrenaline, the neuromodulatory action of CRHR1 described in our study presents different and longer activation kinetics, modulating cell excitability and function through initiation of intracellular signalling without directly influencing membrane potential (see also the new data included in Fig. 6e-h and Supplementary Fig. 7c in our revised manuscript and presented later in this rebuttal). Such actions are long-lasting, compared to classical neurotransmitters and modulators that directly influence membrane potential with fast kinetics. For this reason, without additional measurements – such as concomitant GRAB_{CRH} recordings or calcium recordings, as performed in Silverman et al., 2025⁸ – it is difficult to assess the time scale that a brief optogenetic pulse may have on the extent of CRHR1 activation and its downstream effects. Therefore, we have decided to encapsulate both phases of sigma and perform the stimulation for 50 s, aiming at boosting/prolonging the endogenous CRH release to induce a gain-of-function using ChR2 (or at dampening with PPO). Our stimulation is periodically applied with the closed-loop paradigm during NREMS for a total recording session of ~4h. Fig. 4b and Fig. 5c show portions of such recordings with the colored bars indicating the occurrence of the optogenetic stimulation. The analysis is based on averages across all NREMS bouts during the 4-h stimulation recordings and compared to a matched time window averaged over the two baseline days preceding stimulation. We did not use a same-day pre/post stimulation paradigm. The effects of CRH persisted throughout the recording session: comparing the first and last 2 hours of the session revealed no significant differences in the CRH-influenced measurements (e.g.: no significant change in the average nr of MAs/h of NREM sleep: 44.7 ± 1.09 SEM in the first 2h of the ChR2 stimulation vs. 44.08 ± 1.58 SEM in the last 2h of stimulation, paired t-test p = 0.84).

The parameters used in our study for frequency and intensity of photostimulation were selected based on the known properties of the opsins used, with the goal of avoiding phototoxicity and other off-target effect⁹⁻¹¹. The endogenous firing frequency of CRH neurons projecting to the TRN during NREMS remains unknown. Determining this would help refine stimulation parameters to better mimic physiological activity. The inhibitory opsin PPO¹⁰ is efficiently and transiently activated by blue light at 10 Hz with a 10 ms pulse width, leading to presynaptic inhibition. Accordingly, we used these same parameters for PPO-mediated photoinhibition to ensure reliable opsin activation. For consistency, the same frequency was maintained for ChR2, however, the pulse width was halved because it is known that lower frequencies and shorter pulses are more efficient in activating ChR2 and overstimulation can induce cell toxicity^{9,11}. Furthermore, as mentioned by the Reviewer as well, frequency and stimulation type/duration are known to play a major role in the resulting effects on cellular excitability^{12,13}, and on the behavioural response during sleep¹⁴. Therefore, we chose a stimulation

frequency that we could consistently use across opsins and which induced a sustained behavioural response without inducing cell toxicity.

To further address the Reviewer’s comments here, in a subset of CRH-IRES-Cre × Ai27D mice, we have tested the effect of a 20 Hz stimulation (see figure below). Compared to 10 Hz, this did not consistently produce larger effects on sleep parameters (for instance, it increased microarousals but did not alter sigma power), suggesting that 10 Hz may already be near a ceiling effect. Although testing lower frequencies may also be informative—as shown by Carter et al., 2010¹⁴ in the noradrenergic system—we believe that a proper refinement of the stimulation frequency would require a more extensive work, as it should be based on the knowledge of the endogenous firing patterns of distinct CRH projections. This could allow for more targeted modulation, as different projections may exert differential effects that are not distinguishable in our current setup.

In reply to the Reviewer’s suggestion, we have also added the delta dynamics in the raw data traces of Fig. 4 and 5. The relevant panels from these figures are included below.

Reviewer #2: Similarly, the results from the supplementary Figure 2 on the effects of the optogenetic manipulations can be best answered with longer stimulation protocols. Without such control experiments, the authors might arrive at contrasting results such as those in lines 153-154 (reduction of sigma without affecting its periodicity). These contrasts might be an artifact of the protocol.

Authors' reply: As detailed in the previous response, we selected the photostimulation paradigm in order to encapsulate both phases of sigma, while avoiding exhaustion of CRH release.

Regarding the sigma infraslow analysis presented in the revised Supplementary Fig. 3f, g: these plots demonstrate that the 0.02 Hz periodicity is preserved, e.g., this happens when boosting CRH release with photostimulation, despite an overall reduction in sigma power that we show in Fig. 4d. This preservation of periodicity is consistent with the understanding that sigma infraslow oscillations are governed by mechanisms that are likely independent of CRH—such as noradrenaline release—even though the exact mechanisms are not yet fully elucidated. Therefore, we do not view the preserved periodicity alongside a reduced %sigma power as a contradiction; rather, it is consistent with the view that CRH modulates the ability of TRN to sustain repetitive bursting (thus affecting global sigma power), without preventing bursting initiation, thus maintaining the sigma periodicity.

Reviewer #2: Sigma power as a proxy for spindle density: Although sigma power is among the best representations of spindle density, the specific detection of sleep spindles and quantification of spindle density can strengthen the study's conclusions (as in line 159).

Authors' reply: We fully agree with the Reviewer that specific quantification at the level of sleep spindles would be highly beneficial to understand in more detail the effects of CRH on sleep regulation. However, spindle detection from skull surface electrodes EEG in mice (as the ones we have used in our study) is currently not appropriate or consistent for accurate analysis of sleep spindles. Instead, cortical LFPs are traditionally used for spindle detection in mice. Follow-up studies expanding on our initial results by using other *in vivo* electrophysiological methods combined with CRH measures or calcium recordings would greatly enrich the detail necessary to further understand the role of CRH in sleep regulation.

Reviewer #2: Corelease glutamate and CRH: One of the major results is the corelease of glutamate with the CRH terminals stimulation. Intriguingly, glutamate in PV cells does not affect sleep, as shown in the results in Figure 7 i-k. What are the author's interpretations of this result? Would manipulation of glutamate signaling be sufficient or necessary to modulate any of these responses? Although evidently out of the scope of the study, it would be at least recommended that the discussion on the matter be strengthened. Alternatively, if time allows, pharmacological/optogenetic manipulations of such pathways could significantly increase the significance of the study.

Authors' reply: Indeed, upon wide-field LED illumination in slice recordings, approximately 50% of recorded TRN neurons exhibited excitatory responses time-locked to the LED flashes (revised Supplementary Fig. 8b), suggesting that a subset of TRN cells receive functional glutamatergic input from CRH-expressing fibers. Previous work has reported glutamatergic projections from the basolateral amygdala (BLA) to the TRN¹⁵, however, CRH-releasing neurons from BLA to the nucleus accumbens have been described as GABAergic⁵, as well as general mixed long-range projections to thalamic relays¹⁶. Zona incerta (ZI) is predominantly GABAergic, although it also contains glutamatergic neurons that may innervate the TRN¹⁷. Additionally, a recent study also describes CRH neurons within 'larger' neurons in the medial geniculate nucleus (MGM)¹⁸, which are primarily glutamatergic¹⁹. These data point to three candidate sources for CRH-glutamate co-release. We agree with the Reviewer that further investigation into the identity and functional role of these glutamatergic inputs is important.

However, as rightly noted, such an analysis would require substantial additional experimentation beyond the scope of the current study.

Importantly, the heterogeneity of glutamatergic synapses onto TRN neurons appears to be critical for shaping spindle dynamics. For instance, cortico-TRN synapses are characterized by high expression of GluA4-containing AMPARs and GluN2C-containing NMDARs, which facilitate TRN bursting initiation. In contrast, thalamocortical (TC)-TRN synapses, enriched in GluN2B-containing NMDARs, are more involved in synchronizing spindle activity (reviewed in Fernandez et al., 2020²⁰). Therefore, precise identification of both the presynaptic source and postsynaptic receptor composition of glutamatergic inputs is essential to understanding how CRH-fiber-mediated excitation influences TRN rhythmogenesis. As a result, a general pharmacological approach targeting excitatory transmission would be insufficient to disentangle the specific contributions of distinct glutamatergic synapses onto TRN neurons.

To begin assessing the functional impact of CRH-fiber glutamatergic input on TRN excitability, in this revision of our manuscript we have now conducted additional experiments comparing photo-evoked EPSPs in CRH-IRES-Cre x Ai27D mice to those in Ntsr1-Cre x Ai27D mice, in which Chr2 is selectively expressed in layer 6 cortical neurons. This allowed us to selectively activate cortical glutamatergic inputs using comparable genetic tools and stimulation parameters. As shown in revised Supplementary Fig. 8c (also pasted below), EPSPs evoked by CRH-releasing fiber activation were significantly smaller and did not induce TRN bursting. In contrast, cortical inputs reliably produced larger EPSPs that summated effectively to trigger low-threshold bursts. Please note that the pulse duration of 1-5 ms we used in CRH-IRES-Cre x Ai27D mice was eliciting suprathreshold responses even with a single stimulus in Ntsr1-Cre X Ai27D mice. For this reason, we reduced from the LED flash to 100 μ s, which allowed us to record subthreshold EPSPs in Ntsr1-Cre X Ai27D mice and to compare the peak amplitude between genotypes. These data suggest that glutamate co-released with CRH may have minimal influence on TRN sleep rhythmogenesis, consistent with the lack of impact on the sleep profile when activating CRH-afferents in with shRNA-downregulated CRHR1.

c Comparison of EPSPs evoked in TRN neurons by photostimulation in CRH-IRES-Cre X Ai27D mice and Ntsr1 X Ai27D mice. Left, examples from three different neurons in each mouse line. Spikes elicited by low-threshold bursting in Ntsr1 X Ai27D mice are clipped. LED flashes had a duration of 1-5 ms in CRH-IRES-Cre X Ai27D mice and 100 μ s in Ntsr1 X Ai27D mice. Right, quantification of EPSP peak amplitude using different LED power intensities. For CRH-IRES-Cre X Ai27D mice, only recordings with time-locked responses are included.

We have extended our discussion to include the findings reported in revised Supplementary Fig. 8c:

The complete loss of effects on sleep fragmentation in CRHR1 knockdown mice supports the conclusion that these effects are primarily mediated by CRHR1 activation in the TRN, independent of other synaptic neurotransmitters or neuromodulators that may be co-released with CRH. Our data suggest that CRH afferents to the TRN can also release glutamate, at least onto a subset of TRN neurons. Although this excitatory input does not appear to influence TRN sleep rhythmogenesis, these findings underscore the need for a more detailed investigation into the chemical identity of the CRH-releasing inputs. Future studies should aim to characterize the endogenous activity patterns of

individual CRH projection pathways during sleep and determine their complete neurochemical profile to clarify their specific roles in sleep regulation.

Reviewer #2: Similarly, what effects does it have on the CRHR1 KO in the TRN? Is sleep rhythms, architecture, or fragmentation affected in any way by this manipulation? Comparing the baseline condition of Animals in Figure 7h-k to the baseline of their control group would be sufficient to support/contrast the claims from the results in Figure 5 on CRH inhibition during NREM sleep.

Authors' reply: We have conducted additional analyses to compare baseline sleep between CRHR1 KD and control mice. The sample size has been expanded to include one additional animal that was excluded from the optogenetic study due to optic fiber misplacement. The results, presented in Supplementary Fig. 9 (pasted below), show no differences in sleep architecture (panels a–c). Interestingly, sigma power was lower in CRHR1 KD mice (panel d), which may reflect a compensatory regulation of TRN rhythmogenesis in response to chronic loss of CRHR1 signaling. Nevertheless, we believe this observation does not compromise the conclusions drawn from the optogenetic study, where the assessment of the effects of acute CRH release is performed by comparing baseline vs. stimulated condition within subjects.

Supplementary Fig. 9. CRHR1 downregulation in TRN does not cause major sleep alterations

a-c CRHR1 downregulation in TRN via shRNAmir virus used to knock-down (KD) CRHR1 did not alter the number of MAs per h of NREMS (MAs, **a**), the % of NREMS (**b**) or the length of NREMS bouts (**c**). **d, e** CRHR1 downregulation in TRN decreased sigma power (**d**), but did not affect delta power (**e**).

Reviewer #2: Cellular mechanisms of CRHR1 on calcium burst: From a nonexpert perspective on in-vitro electrophysiology, the outstanding additional explanatory level of the cellular mechanisms represents a major contribution to the field. I am also curious about the underlying mechanisms of suppressing calcium bursting in TRN neurons. Is it via partial depolarization of the TRN cells or by internal modulation of the Ca_v3 Ca²⁺? A more detailed explanation or voltage clamp examples of such effects might promote and facilitate further discussions in the field.

Authors' reply: To address the Reviewer's comment, we have now substantially expanded our electrophysiological characterization *ex vivo* to explore the cellular basis of CRH-mediated inhibition of TRN bursting. Specifically, we performed patch-clamp recordings to assess the effects of CRH on isolated Ca_v3 and SK2 currents, which are key contributors to rhythmic bursting in TRN neurons. As shown in the revised manuscript (Fig. 6e-h; see these figure panels pasted below), our results demonstrate that CRH significantly reduces Ca_v3-mediated Ca²⁺ currents. We further report that CRH

does not depolarize TRN cells (Supplemental Figure 7b). While we cannot entirely rule out the involvement of other ionic mechanisms, these findings provide a mechanistic explanation for the influence of CRH on spindle rhythmogenesis.

e Left, example traces of a voltage-clamp recording showing mean biphasic currents elicited in a TRN neuron before (Bsl) and after (CRH) bath-application of 500 nM CRH. The relative contribution of Ca_{v3}-channel- and SK2-channel-mediated currents to the total charge transfer is indicated by the color-coded areas. Right, bath-applied CRH did not alter the positive portion of the charge transfer, but induced a slight decrease in the negative charge ($p = 0.076$), which was absent in control recordings without CRH application ($n = 7$ for Ctr, $n = 8$ for CRH).

f Left, example traces of a voltage-clamp recording showing mean SK2-channel-mediated tail currents elicited in a TRN neuron at the offset of a brief depolarization to +20 mV —activating high voltage-gated Ca²⁺ channels— before (Bsl) and after (CRH) bath-application of 500 nM CRH. The current response to the depolarizing step is cut. The analysed portion of the tail current is color-coded. Right, time course of the percentage change of the charge transfer and the decay time of the tail current, indicating no significant impact of CRH on the SK2-mediated tail current ($n = 7$, $p > 0.05$).

g Left, example traces of a voltage-clamp recording showing mean isolated Ca_{v3} currents elicited in a TRN neuron before (Bsl) and after (CRH) bath-application of 500 nM CRH. The inset shows a magnified portion around the current peak. Right, time course of the percentage change of Ca_{v3} current peaks, indicating that CRH (500 nM, $n = 9$) induced a decrease that was prevented by the CRHR1 antagonist NBI35965 (NBI, 3 μ M, $n = 6$).

h Left, example traces of isolated Ca_{v3} currents elicited in a TRN neuron by repetitive depolarizations before (Bsl) and after (CRH) bath-application of 500 nM CRH. Right, mean percentage change of Ca_{v3} current peaks ($n = 5$), indicating a progressive increase in the effect of CRH.

In our revised manuscript, we describe these results as follows:

Next, we investigated the ionic mechanisms underlying CRH-induced burst reduction. In current-clamp experiments, bath application of CRH did not alter the membrane potential (Supplementary Fig. 7b), consistent with the reported modulatory actions of CRHR1²¹, indicating that the shift in burst activation was not due to neuronal hyperpolarization or depolarization. Next, we conducted voltage-clamp experiments to analyze the primary ionic currents sustaining cyclic bursting—namely, Ca_{v3}-mediated Ca²⁺ currents and SK2-mediated currents^{22–24}. In TRN cells patched with a K⁺-methyl sulfate-based intracellular solution in the presence of the Na⁺ channel blocker tetrodotoxin, we examined biphasic currents elicited by 40-mV depolarizations from a holding potential of -75 mV. These consisted of an inward Ca_{v3}-mediated Ca²⁺ component that was curtailed by an outward SK-mediated current (Fig.

6e). To obtain an initial assessment of the effect of CRH on these currents, we measured the charge transfer associated with the biphasic current. Since the negative phase primarily reflects Ca_v3 -mediated depolarizing Ca^{2+} influx, a reduction in Ca_v3 channel activity would be expected to decrease this component. Conversely, SK2 channel modulation would primarily affect the positive phase, representing repolarizing K^+ efflux²⁴. Bath application of CRH induced a trend toward a reduction in the negative charge component ($p = 0.076$) without affecting the positive charge (Fig. 6e). Control recordings without CRH application indicate no change in either component. The CRH-induced reduction in the negative charge suggests that CRH may affect the Ca_v3 contribution, although the extent of the effect could be masked by the tight coupling between Ca_v3 and SK2 currents. To further dissect the individual contributions of these conductances, we modified our recording conditions. To isolate SK2 currents from Ca_v3 -mediated Ca^{2+} influx, we leveraged the fact that SK2 currents can also be triggered by Ca^{2+} influx through high-voltage-gated Ca^{2+} channels. In TRN cells held at -50 mV, brief depolarizations to $+20$ mV elicited a tail current (Fig. 6f), which is mediated by SK2 channels^{23,24}. Neither the charge nor the decay time of this tail current was affected by bath application of CRH (Fig. 6f), suggesting no significant impact of CRH on SK2 currents. To isolate Ca_v3 currents, we switched to a Cs^+ -gluconate-based intracellular solution to block all K^+ channels, including SK2. Under these conditions, low-voltage-activated Ca_v3 currents were significantly reduced by CRH, an effect that was prevented by the CRHR1 antagonist NBI35965 (Fig. 6g). CRH decreased the peak amplitude of Ca_v3 currents by approximately 10%, raising the question of whether this modest effect was sufficient to explain the observed reduction in cyclic TRN cell bursting. To test this, we elicited repetitive Ca_v3 currents by applying three consecutive brief depolarizations (Fig. 6h). Notably, CRH had a more pronounced effect on Ca_v3 currents after the initial peak, consistent with its reported influence on Ca_v3 channel recovery from inactivation²⁵. This finding on progressive attenuation of Ca_v3 currents aligns with the dampening of repetitive bursting observed in TRN cells upon CRHR1 activation.

Reviewer #2: Sleep stages and power spectra: Representative epochs of the sleep stages and average (or grand average) power spectra for each sleep stage (with the different manipulations) and not only the average sigma and/or delta power are often ideal to represent the effects in overall brain activity. Supplementary information containing such figures gives the reader additional support for the quality and strength of the results. These can be done in log-linear or log-log spectra, which might also be enough to support your claims to skeptical readers.

Authors' reply: In response to the Reviewer's suggestion, we have included example EEG/EMG traces and mean power spectra (log-linear) for NREMS and REMS during baseline and stimulation session in the revised Supplementary Fig. 3 and Supplementary Fig. 5. The new panels and the corresponding figure legends are pasted below.

Supplementary Fig. 3. Optogenetic stimulation CRH release affects NREMS architecture but not sigma infraslow periodicity

a, b Example portions of EEG/EMG traces (with enlarged scales in the insets) and NREMS and REMS power spectra during baseline (bsl) and stimulation (stim) session in a CRH-IRES-Cre X Ai27D mouse

Supplementary Fig. 5. Optogenetic inhibition of CRH release affects NREMS architecture but not sigma infraslow periodicity

a, b Example portions of EEG/EMG traces and NREMS and REMS power spectra during baseline (bsl) and stimulation (stim) session in a CRH-IRES-Cre mouse expressing Parapinopsin (PPO) in CHR-releasing neuron targeting TRN.

Reviewer #2: REM sleep transitions: Although the latency to REM sleep is an appropriate proxy for the CRH effect on NREM-to-REM transitions, the most convincing metric to support some of the conclusions is the actual NREM-to-REM transition density per [unit of time (min or hr)] of NREM sleep. Additionally, the optogenetics protocols raise the question of how much the NREM-to-REM transition latency effects depend on a post-manipulation rebound. An average transition density before, during, and after the optogenetic manipulation(s) might help clarify.

Authors' reply: In response to the Reviewer's suggestion, we have now calculated the density of NREMS-to-REMS transitions (TTRs), reported as the number of TTRs per hour of NREMS, for both baseline and stimulation conditions. These data are presented in the Supplementary Fig. 3h and 5h shown below. The TTR density remains unchanged with Chr2 stimulation, but shows a significant decrease with PPO stimulation ($p = 0.03$, paired t-test), supporting the view that CRH inhibition consolidates NREMS episodes. It is important to note that a post-manipulation rebound cannot be reliably assessed in our experimental setup. Due to the kinetics of CRHR1 signaling and the nature of our stimulation protocol—repeated stimulation across the entire recording period—we do not analyze same-day pre/post effects. Moreover, most TTRs occur during the 50-second stimulation window, which is triggered by a peak in sigma power following the rule of the closed-loop stimulation. Given that a typical TTR in mice often begins before the sigma peak and lasts around 50 seconds, REMS frequently initiates before the end of the stimulation window, thus excluding post-stimulation rebound.

h Photostimulation did not change the number of transitions to REMS per hour of NREMS.

h Photoinhibition decreased the number of transitions to REMS per hour of NREMS

Reviewer #2: Similarly, what is the rationale behind the pre-REM sigma peak analyses of Figures 4f and 5g? Given the result of reduced averaged sigma activity, these accessory results seem redundant or do not seem to provide additional information. Instead, they raise more questions about the stimulation protocol's effect and post-manipulation(s) rebound. It's important to clarify the relevance of these results and how they contribute to the study's conclusions.

Authors' reply:

We respectfully disagree with the Reviewer's assessment that the analysis of sigma power at transitions is redundant or merely accessory. The period preceding REMS represents a distinct sleep stage, often referred to as intermediate sleep, which is conserved across species. This stage is characterized by a surge in spindle density, leading to a local maximum in infraslow sigma oscillations. It coincides with a unique neuromodulatory environment, involving concurrent noradrenergic and cholinergic activity, which facilitates both increased spindle generation and the emergence of REM-like hippocampal theta activity^{20,26}.

While our findings do not support a direct induction or suppression of TTRs or REM sleep by CRH—as it has been reported for noradrenaline acting on TRN neurons during NREMS—we do observe changes in sigma power dynamics at these transitions. Specifically, although TTRs often occur during the stimulation window (when CRH levels are typically decreasing, as shown by our GRAB_{CRH} recordings), we find a general reduction in sigma power and an increased latency between the sigma peak and REMS onset. These alterations in TTR dynamics are consistent with recent human studies (e.g., Wassing et al., 2019²⁷), where a reduced sigma peak or prolonged TTR duration have been associated with impaired emotional adaptation, particularly involving the amygdala, during subsequent REM sleep.

Therefore, our view is that this analysis provides meaningful insight into how CRH may modulate the quality and structure of REMS transitions in ways that could be relevant to stress- and anxiety-related disorders. This interpretation was already included in the Discussion section (pasted below), as we believe it is an important and clinically relevant avenue for future investigation.

“Increasing CRH release with photostimulation also reduced sigma surge at the transition to REMS and delayed REMS onset, mirroring the increased REMS latency observed in stress- and anxiety-related disorders such as generalized anxiety and PTSD^{28,29}. In humans, reduced spindle density at the transition to REMS disrupts the overnight amygdala adaptation during REMS, leading to daytime hyperarousal seen in various psychiatric disorders²⁷. These findings suggest that CRH may contribute to ‘restless REMS’ and related disorders. Investigating whether anxiety-inducing events during wakefulness alter CRH dynamics during subsequent sleep could provide insights into how emotional experiences influence the neuromodulatory tone during NREMS and impact limbic adaptation during REMS.”

Reviewer #2: Finally, how much does the minimum bout length of REM sleep (5 epochs, 20 s) affect the NREM-to-REM transition result? Would it be the same for shorter (2 to 3 epochs)?

Authors’ reply: We have performed the analysis for TTRs using a shorter (2 epochs of REMs) threshold for detecting the TTR. The results (revised Supplementary Fig. 3i-k, included below) do not vary with the length of the detection threshold, indicating no bias in the observation. In general, we have set as ‘consolidated’ bouts the ones of at least 5 epochs (20 s) unless otherwise stated, which is the reason the same number was used for the detection of consolidated REMS as well as NREMS.

i-k Analysis of NREMS-REMS transitions using a minimum bout length = 2 for REMS episode detection confirms the results found with minimum bout length = 5, i.e., significant reduction in sigma peak and increased latency to transition upon photostimulation.

Reviewer #2: Minor comments

- In lines 56-57: If needed, the statement “Notably, 56 NREMS fragility periods have a higher probability for the occurrence of MAs” might be supported by Cardis et al., 2021.

Authors' reply: We thank the Reviewer for noticing this oversight. Indeed, the statement should be followed by citations of Cardis et al., 2021 and Lecci et al., 2017^{30,31}. We have implemented this change.

Reviewer #2: In line 446, AP, ML, and DV for the coordinates are missing.

Authors' reply: This has been fixed.

- Figure titles sometimes refer to the result and other times to the methods. As in the case of the results, one suggestion is to homogenize one approach or another for the figures.

Authors' reply: We have reviewed all figure titles and revised those in Supplementary Figures 3 and 8 to more clearly reflect the results presented, ensuring greater consistency across the figures.

Reviewer #2: - In Figure 3e, grey lines represent the peaks and troughs of the sigma activity, yet they are not described in the legend.

- Figure 3 misses the legend for 3d.

Authors' reply: We thank the Reviewer for noticing the inconsistencies in the legend for fig 3. We have now included the description for 3d and the description of the grey lines for 3e in the legend of Fig 3; text pasted here below:

“d. Power spectral density analysis of CRH release during NREMS shows an overlapping peak with the infraslow oscillation of sigma centered around 0.02 Hz.

e Example trace of the time-matched CRH signal and the infraslow oscillation of sigma during NREMS. Continuous vertical grey lines indicate sigma peaks and dotted vertical grey lines indicate sigma valleys.”

Reviewer #2: For the cross-correlation analysis, it is often helpful to also depict the lag of the peak correlation and not only the correlation at zero lag.

Authors' reply: To address this point, we have now completed Fig. 3f (pasted below) with the distribution of the cross-correlation r value for each mouse at max correlation and with the distribution of the lag values at max correlation.

f Left, cross-correlation of the two signals shows that they anti-correlate, with maximal CRH release during NREMS overlapping with periods of low spindle activity corresponding to sleep fragility ($n = 12$). Right, the upper violin plots show the individual variability of the cross-correlation, each dot is the correlation coefficient (r) at lag 0 (left) and at the maximum correlation (max corr, right) for each mouse; the lower violin plot shows the distribution of the lag values at maximum correlation (lag at max corr).

Reviewer #2: For the list of features in lines 581-587, I suggest using “;” instead of “-” to separate the features, as sometimes “-” could be mistaken for a minus, disrupting the reading fluidity.

Authors' reply:

Thank you for the suggestion. We have modified the interpunction to avoid any ambiguity.

References

1. Wang, H. *et al.* A tool kit of highly selective and sensitive genetically encoded neuropeptide sensors. *Science (1979)* **382**, (2023).
2. Jiang, Z., Rajamanickam, S. & Justice, N. J. Local corticotropin-releasing factor signaling in the hypothalamic paraventricular nucleus. *Journal of Neuroscience* **38**, 1874–1890 (2018).
3. Tervo, D. G. R. *et al.* A Designer AAV Variant Permits Efficient Retrograde Access to Projection Neurons. *Neuron* **92**, 372–382 (2016).
4. Walker, L. C., Cornish, L. C., Lawrence, A. J. & Campbell, E. J. The effect of acute or repeated stress on the corticotropin releasing factor system in the CRH-IRES-Cre mouse: A validation study. *Neuropharmacology* **154**, 96–106 (2019).
5. Birnie, M. T. *et al.* Stress-induced plasticity of a CRH/GABA projection disrupts reward behaviors in mice. *Nat Commun* **14**, (2023).
6. Alon, T. *et al.* Transgenic mice expressing green fluorescent protein under the control of the corticotropin-releasing hormone promoter. *Endocrinology* **150**, 5626–5632 (2009).
7. Osorio-Forero, A. *et al.* Noradrenergic circuit control of non-REM sleep substates. *Current Biology* **31**, 5009-5023.e7 (2021).
8. Silverman, D. *et al.* Activation of Locus Coeruleus Noradrenergic Neurons Rapidly Drives Homeostatic Sleep Pressure. *Sci. Adv* vol. 11 <https://www.science.org> (2025).
9. Boyden, E. S., Zhang, F., Bamberg, E., Nagel, G. & Deisseroth, K. Millisecond-timescale, genetically targeted optical control of neural activity. *Nat Neurosci* **8**, 1263–1268 (2005).
10. Copits, B. A. *et al.* A photoswitchable GPCR-based opsin for presynaptic inhibition. *Neuron* **109**, 1791-1809.e11 (2021).
11. Zhang, F., Wang, L. P., Boyden, E. S. & Deisseroth, K. Channelrhodopsin-2 and optical control of excitable cells. *Nat Methods* **3**, 785–792 (2006).
12. Adamantidis, A. R., Zhang, F., Aravanis, A. M., Deisseroth, K. & De Lecea, L. Neural substrates of awakening probed with optogenetic control of hypocretin neurons. *Nature* **450**, 420–424 (2007).
13. Thankachan, S. *et al.* Thalamic Reticular Nucleus Parvalbumin Neurons Regulate Sleep Spindles and Electrophysiological Aspects of Schizophrenia in Mice. *Sci Rep* 1–16 (2019) doi:10.1038/s41598-019-40398-9.
14. Carter, M. E. *et al.* Tuning arousal with optogenetic modulation of locus coeruleus neurons. *Nat Neurosci* **13**, 1526–1535 (2010).
15. Aizenberg, M. *et al.* Projection from the Amygdala to the Thalamic Reticular Nucleus Amplifies Cortical Sound Responses. *Cell Rep* **28**, 605-615.e4 (2019).
16. Ahmed, N. & Paré, D. The Basolateral Amygdala Sends a Mixed (GABAergic and Glutamatergic) Projection to the Mediodorsal Thalamic Nucleus. *Journal of Neuroscience* **43**, 2104–2115 (2023).
17. Heise, C. E. & Mitrofanis, J. Evidence for a Glutamatergic Projection from the Zona Incerta to the Basal Ganglia of Rats. *Journal of Comparative Neurology* **468**, 482–495 (2004).

18. Tomioka, R., Takemoto, M. & Song, W. J. Neurochemical properties for defining subdivisions of the mouse medial geniculate body. *Hear Res* **431**, (2023).
19. Crabtree, J. W. Functional diversity of thalamic reticular subnetworks. *Front Syst Neurosci* **12**, 1–18 (2018).
20. Fernandez, L. M. J. & Lüthi, A. Sleep Spindles: Mechanisms and Functions. *Physiol Rev* **100**, 805–868 (2020).
21. Deussing, J. M. & Chen, A. The corticotropin-releasing factor family: Physiology of the stress response. *Physiol Rev* **98**, 2225–2286 (2018).
22. Pellegrini, C., Lecci, S., Lüthi, A. & Astori, S. Suppression of Sleep Spindle Rhythmogenesis in Mice with Deletion of $Ca_v3.2$ and $Ca_v3.3$ T-type Ca^{2+} Channels. *Sleep* **39**, 875–885 (2016).
23. Astori, S. *et al.* The $Ca_v3.3$ calcium channel is the major sleep spindle pacemaker in thalamus. **108**, 13823–13828 (2011).
24. Cueni, L. *et al.* T-type Ca^{2+} channels, SK2 channels and SERCAs gate sleep-related oscillations in thalamic dendrites. *Nat Neurosci* **11**, 683–692 (2008).
25. Tao, J. *et al.* Activation of corticotropin-releasing factor receptor 1 selectively inhibits $Ca_v3.2$ T-type calcium channels. *Mol Pharmacol* **73**, 1596–1609 (2008).
26. Durán, E., Oyanedel, C. N., Niethard, N., Inostroza, M. & Born, J. Sleep stage dynamics in neocortex and hippocampus. *Sleep* **41**, (2018).
27. Wassing, R. *et al.* Restless REM Sleep Impedes Overnight Amygdala Adaptation. *Current Biology* **29**, 2351–2358.e4 (2019).
28. Cabrera, Y. *et al.* Overnight neuronal plasticity and adaptation to emotional distress. *Nat Rev Neurosci* (2024) doi:10.1038/s41583-024-00799-w.
29. Cox, R. C. & Olatunji, B. O. A systematic review of sleep disturbance in anxiety and related disorders. *J Anxiety Disord* **37**, 104–129 (2016).
30. Cardis, R. *et al.* Cortico-autonomic local arousals and heightened somatosensory arousability during NREMS of mice in neuropathic pain. *Elife* **10**, 1–27 (2021).
31. Lecci, S. *et al.* Coordinated infraslow neural and cardiac oscillations mark fragility and offline periods in mammalian sleep. *Sci Adv* **3**, (2017).

Reply to REVIEWERS' COMMENTS

Reviewer #1 (Remarks to the Author):

Thank you to authors for their patience and thorough response in addressing all my concerns. The authors have now added several additional experiments and added methodological details and discussion that improve study and quality of the manuscript. It is now suitable for publication. Congratulations on an excellent study!

Authors' reply

We are pleased that the revised version meets the Reviewer's expectations and are grateful for their support of our study.

Reviewer #2 (Remarks to the Author):

The manuscript by Cumpuna et al. is an excellent example of a hypothesis-driven approach in systems neuroscience, particularly in the field of sleep. They studied the network, cellular, and functional mechanisms of corticotropin-releasing hormone (CRH) in the Reticular Thalamic Nucleus (TRN) and its role in sleep architecture, microarchitecture, and spontaneous arousability in NREM sleep. They described close-to-minute fluctuations in CRH in the TRN, similar to those of other monoaminergic systems, and related to electrophysiological markers of arousability on this timescale. They also demonstrated the sufficiency of this molecule in modulating the fragility of NREM sleep through optogenetic and genetic approaches. Furthermore, through rigorous *ex vivo* experiments, they demonstrated that CRH acts on the spindle-generating PV cells of the TRN by progressively attenuating Cav3 currents. I praise the excellent figures throughout the manuscript, which effectively depict the significant results in a clear and accessible manner, even with the example/representative traces. In this revised version of the manuscript, the authors addressed all my concerns in detail. I do not have further relevant concerns, but I have included my response to some of the raised points below.

Authors' reply:

We thank the Reviewer for the appreciation of our study and the quality of data representation. We greatly appreciate the time and effort dedicated to examining our revised manuscript and for providing additional feedback.

Through their explanation, I am less worried, although still curious, about the lack of effect on the infraslow fluctuations in sigma that accompany the reduction in sigma after optogenetic stimulation. However, in light of the outstanding, long-lasting impact of CRH in TRN cells (as shown in the *ex vivo* experiments), this raises questions on the nature of the sigma activity in comparison with individual spindles or other electrophysiological characteristics of the signal (e.g., aperiodic activity).

Authors' reply:

We are glad that the additional information and rationale provided in the revision helped clarifying the concerns raised by the Reviewer regarding the lack of CRH impact on sigma infraslow periodicity. As outlined by our replies to this and related comments, the use of local field recordings could enable us to detect sleep spindles with higher fidelity than with scalp EEG measures and possibly allow us to dissect the effect of CRH on discrete spindles vs. global sigma activity.

- In this line of thought. I agree that local field potentials (particularly those of the somatosensory cortex) are best suited for spindle analysis. However, the EEG has also been used in the past to study spindles in rodents (e.g., Mölle et al., 2009; Lacroix et al., 2018; Fernandez et al., 2018). Although outside the scope of this study, it would be interesting to address these differences in the future. I share a simple script in this response that may be useful for quantifying spindles in NREM sleep for future reference.

Authors' reply:

We thank the Reviewer for highlighting relevant literature on spindle detection and for sharing a script that may be useful for future analyses. To our understanding, spindle detection in the studies by Lacroix et al. (2018) and Fernandez et al. (2018) was performed on local field potential (LFP) recordings rather than scalp EEG. The study by Mölle et al. (2009), while using EEG, was conducted in rats rather than mice. Detecting spindles in mice using EEG presents specific challenges due to lower signal-to-noise ratios and anatomical constraints. Nonetheless, recent work (e.g., Uygun et al., 2019; doi:10.1093/sleep/zsy218) has begun to address these limitations through automated detection methods. We agree that further exploration of these methodological differences would be valuable and appreciate the Reviewer's input for future directions.

- Concerning the stimulation protocols. I apologize for the lack of precision in my comment regarding the stimulation frequencies, as I was curious about the lower frequencies (not higher). However, first, it is still very valuable to highlight the potential ceiling effect of the 10 Hz stimulation with the new control experiment at 20 Hz. I agree with the authors in emphasizing the complexity and scope of such experiments, which would be interesting to conduct in other studies, particularly those examining specific firing patterns of CRH+ cells. Co-release mechanisms are a fascinating and complex topic worth tackling across neural systems.

Authors' reply:

We acknowledge the Reviewer's point regarding the relevance of testing lower stimulation frequencies. As noted, the role of endogenous CRH+ cell firing adds further complexity to this question, but it represents a promising direction for future research.

- I had a previous concern about what turned out to be one of the most exciting results. Namely, the NREM-to-REM sleep transitions in the *in vivo* manipulations. The creative analysis of REM latency in comparison to the sigma peak, or in addition to the transition density (TTR in the manuscript), highlights the importance of the intermediate state and the complexity of the neurochemical state, particularly in the context mentioned by the authors, namely stress-related disorders.

Authors' reply:

We are happy that the additional information provided in the revision helped clarifying the relevance of our findings on the impact of CRH at NREMS-REMS transitions.

References

- Mölle M, Eschenko O, Gais S, Sara SJ, Born J. The influence of learning on sleep slow oscillations and associated spindles and ripples in humans and rats. ****Eur J Neurosci**** 29: 1071–1081, 2009. doi:10.1111/j.1460-9568.2009.06654.x.
- Lacroix MM, de Lavilléon G, Lefort J, El Kanbi K, Bagur S, Laventure S, Dauvilliers Y, Peyron C, Benchenane K. Improves sleep scoring in mice reveals human-like stages. ****bioRxiv**** 489005, 2018. doi:10.1101/489005.
- Fernandez LM, Vantomme G, Osorio-Forero A, Cardis R, Béard E, Lüthi A. Thalamic reticular control of local sleep in mouse sensory cortex. ****eLife**** 7: e39111, 2018. doi:10.7554/eLife.39111.

```

function st_Output = f_SimpleSpDetection(v_Signals,s_Fs,v_Hyp,s_Threshold)
% f_SimpleSpDetection
% Detects sleep spindles in EEG data using bandpass filtering and amplitude
thresholds,
% based on methods from Fernandez et al., 2018 and Osorio-Forero et al.,
2021, 2024.
%
% Inputs:
%   v_Signals    - Raw EEG signal vector.
%   s_Fs         - Sampling frequency in Hz.
%   v_Hyp       - Hypnogram (same length as v_Signals) with sleep states:
%                 0 = Other, 1 = REM, 2 = NREM, 3 = Wake
%   s_Threshold  - Threshold in standard deviations above NREM mean for
detection.
%
% Output:
%   st_Output - Structure containing:
%       m_Spindles      - 2xN matrix with start and end sample indices
of N detected spindles.
%       v_Sp_Amp       - Maximum amplitude of each spindle.
%       v_Sp_Speed     - Mean frequency (Hz) within each spindle.
%       v_Sp_NCycles   - Number of cycles (peaks) within each spindle.
%       v_Sp_Time      - Duration of each spindle in seconds.
%       v_Sp_Pow       - Total power (sum of squares) within each
spindle.

% -----
% Preprocessing
% -----

% Remove DC offset by subtracting mean
v_Signals = v_Signals - nanmean(v_Signals);

% Ensure signal is a row vector
if size(v_Signals,2) < size(v_Signals,1)
    v_Signals = v_Signals';
end

% Replace NaNs with 0
v_Signals(isnan(v_Signals)) = 0;

% Fix hypnogram boundaries if starting/ending with NREM
if v_Hyp(1) == 2, v_Hyp(1) = 0; end
if v_Hyp(end) == 2, v_Hyp(end) = 0; end

disp('Filtering...')

% -----
% Bandpass filter the signal between 9-16 Hz (spindle band)
% -----
bhi = fir1(2000, [9 16]/(s_Fs/2)); % FIR filter design
v_Filtered = filtfilt(bhi, 1, v_Signals); % Zero-phase filtering
v_FilteredSquared = v_Filtered.^2;

% Find indices corresponding to NREM

```

```

v_InNR = find(v_Hyp == 2);

% -----
% Spindle Detection (based on squared filtered signal)
% -----
tic

% Set amplitude threshold based on NREM mean + s_Threshold * std
s_Threshold = mean(v_FilteredSquared(v_InNR)) + s_Threshold *
std(v_FilteredSquared(v_InNR));

% Find peaks in the squared filtered signal
[v_PeakVal, v_PeakLoc] = findpeaks(v_FilteredSquared);

% Keep peaks that occur during NREM
[v_PeakLoc, idxKeep, ~] = intersect(v_PeakLoc, v_InNR);
v_PeakVal = v_PeakVal(idxKeep);
v_AllPeaksPos = v_PeakLoc;

% Retain only peaks above the amplitude threshold
aboveThresh = v_PeakVal >= s_Threshold;
v_PeakLoc = v_PeakLoc(aboveThresh);
v_PeakVal = v_PeakVal(aboveThresh);

% Find negative peaks (for cycle boundaries)
[~, v_DropsLoc] = findpeaks(-v_FilteredSquared);

% Ensure drop locations span the full range
if v_PeakLoc(1) < v_DropsLoc(1)
    v_DropsLoc = [1, v_DropsLoc];
end

% -----
% Define potential spindles based on inter-peak intervals
% -----
v_TimeBetweenPeaks = diff(v_PeakLoc) / s_Fs;
v_TooLong = find(v_TimeBetweenPeaks > (2*16/s_Fs)); % >2 cycles

% Possible spindle start/end points
v_PosibleEnd = [v_PeakLoc(v_TooLong), v_PeakLoc(end)];
v_PosibleStart = [v_PeakLoc(1), v_PeakLoc(v_TooLong + 1)];

% Refine start/end using nearby drop locations (2 before and after)
v_RealStart = zeros(1, length(v_PosibleStart));
v_RealEnd = zeros(1, length(v_PosibleEnd));
for i = 1:length(v_PosibleStart)
    dropsBefore = find(v_DropsLoc < v_PosibleStart(i), 2, 'last');
    dropsAfter = find(v_DropsLoc > v_PosibleEnd(i), 2, 'first');
    if length(dropsBefore) == 2 && length(dropsAfter) >= 1
        v_RealStart(i) = v_DropsLoc(dropsBefore(end));
        v_RealEnd(i) = v_DropsLoc(dropsAfter(1));
    end
end

% Remove unused preallocated entries
v_RealStart = v_RealStart(v_RealStart > 0);

```

```

v_RealEnd = v_RealEnd(1:length(v_RealStart));

% -----
% Merge nearby spindles (within 50 ms)
% -----
v_TimeBetweenSpindles = diff(v_RealStart) - diff([0, v_RealEnd(1:end-1)]);
v_TimeBetweenSpindles = v_TimeBetweenSpindles / s_Fs;
v_MergeHere = find(v_TimeBetweenSpindles < 0.05); % <50 ms
v_RealEnd(v_MergeHere) = [];
v_RealStart(v_MergeHere+1) = [];

% -----
% Final validation: keep spindles with 9â€"16 Hz and â¥3 cycles
% -----
idxLF = 1;
while idxLF <= length(v_RealEnd)
    v_PeaksInSp = find(v_AllPeaksPos > v_RealStart(idxLF) & v_AllPeaksPos <
v_RealEnd(idxLF));
    if length(v_PeaksInSp) > 5
        peakDiffs = diff(v_AllPeaksPos(v_PeaksInSp)) / s_Fs;
        freq = mean(1 ./ peakDiffs);
        if freq < 18 && freq > 9
            idxLF = idxLF + 1;
        else
            v_RealStart(idxLF) = [];
            v_RealEnd(idxLF) = [];
        end
    else
        v_RealStart(idxLF) = [];
        v_RealEnd(idxLF) = [];
    end
end

% -----
% Mark final spindles in output
% -----
m_Spindles = [v_RealStart; v_RealEnd];

% -----
% Extract spindle features
% -----
for i = 1:length(m_Spindles)
    idx1 = m_Spindles(1, i);
    idx2 = m_Spindles(2, i);
    spindle = v_Filtered(idx1:idx2) - mean(v_Filtered(idx1:idx2));

    v_Sp_Amp(i) = max(abs(spindle)); % Peak amplitude
    [~, posPeaks] = findpeaks(spindle);

    v_Sp_NCycles(i) = length(posPeaks); % Number of peaks
    cycleDur = diff(posPeaks) / s_Fs;
    v_Sp_Speed(i) = 1 / mean(cycleDur); % Frequency (Hz)
    v_Sp_Time(i) = (idx2 - idx1) / s_Fs; % Duration in seconds
    v_Sp_Pow(i) = sum(spindle .^ 2); % Power
    v_SpindlesLoc(i) = round((idx1 + idx2) / 2); % Midpoint location
end

```

```

% -----
% Continuity of spindles to check if they come in clusters, this is not
% saved but if you want to save it change it in the st_output bellow
% -----
%
%     v_Temp = diff(v_Hyp==2);
%     m_NRboutsOrg = [find(v_Temp==1);find(v_Temp==-1)];
%
%     v_Temp = (m_NRboutsOrg(2,:)-m_NRboutsOrg(1,:))/10;
%     m_NRboutsOrg(:,v_Temp<100)=[];
%
%     v_TempLocBinary = zeros(1,length(v_Signals));
%     v_TempLocBinary(v_SpindlesLoc) = 1;
%     v_AllDistances = [];
%     for idxBout = 1 : length(m_NRboutsOrg)
%         v_TempTime =
v_TempLocBinary(m_NRboutsOrg(1,idxBout):m_NRboutsOrg(2,idxBout));
%         v_LocTimes = find(v_TempTime==1);
%         v_LocTimes = (v_LocTimes(2:end)-v_LocTimes(1:end-1))/s_Fs;
%         v_AllDistances = [v_AllDistances,v_LocTimes];
%     end

% -----
% Save output
% -----
st_Output.m_SpindlesIn200Hz = m_Spindles;
st_Output.v_Sp_Amp          = v_Sp_Amp;
st_Output.v_Sp_Speed       = v_Sp_Speed;
st_Output.v_Sp_NCycles     = v_Sp_NCycles;
st_Output.v_Sp_Time        = v_Sp_Time;
st_Output.v_Sp_Pow         = v_Sp_Pow;

% Optional: v_Sp_InterDistance, not computed by default
% Uncomment relevant sections if needed

disp(['Time: ', num2str(toc), ' sec']);

en

```